# Think Only When You Need
# with Large Hybrid-Reasoning Models

**Lingjie Jiang**[*†‡]  **Xun Wu**[*†]  **Shaohan Huang**[†]  **Qingxiu Dong**[†‡]  **Zewen Chi**[†]
**Li Dong**[†]  **Xingxing Zhang**[†]  **Tengchao Lv**[†]  **Lei Cui**[†]  **Furu Wei**[†◇]
[†] Microsoft Research    [‡] Peking University
lingjiejiang@stu.pku.edu.cn; {xunwu, shaohanh, fuwei}@microsoft.com
https://aka.ms/GeneralAI

## Abstract

Recent Large Reasoning Models (LRMs) have shown substantially improved reasoning capabilities over traditional Large Language Models (LLMs) by incorporating extended thinking processes prior to producing final responses. However, excessively lengthy thinking introduces substantial overhead in terms of token consumption and latency, particularly unnecessary for simple queries. In this work, we introduce Large Hybrid-Reasoning Models (LHRMs), the first kind of model capable of adaptively determining whether to perform thinking based on the contextual information of user queries. To achieve this, we propose a two-stage training pipeline comprising Hybrid Fine-Tuning (HFT) as a cold start, followed by online reinforcement learning with the proposed Hybrid Group Policy Optimization (HGPO) to implicitly learn to select the appropriate thinking mode. Furthermore, we introduce a metric called Hybrid Accuracy to quantitatively assess the model's capability for hybrid thinking. Extensive experimental results show that LHRMs can adaptively perform hybrid thinking on queries of varying difficulty and type. It outperforms existing LRMs and LLMs in reasoning and general capabilities while significantly improving efficiency. Together, our work advocates for a reconsideration of the appropriate use of extended thinking processes, and provides a solid starting point for building hybrid thinking systems.

## 1  Introduction

Recent remarkable advancements in Large Reasoning Models (LRMs), such as DeepSeek-R1 [15], OpenAI o1/o3 series [38] and others [46, 37], have catalyzed a significant paradigm shift from conventional Large Language Models (LLMs) toward LRMs. Compared to LLMs, LRMs exhibit substantially improved reasoning capabilities and generalization across tasks such as programming, mathematics, and commonsense reasoning [49, 15, 48]. These improvements are primarily attributed to the generation of extended thinking traces, often marked by special tokens such as `<think>`, prior to producing the final responses. Despite the effectiveness of LRMs, existing work primarily focuses on converting LLMs into LRMs to enhance performance across various domains, typically through reinforcement learning algorithms such as GRPO [15] and REINFORCE++ [17], or by fine-tuning on distilled thinking traces [36].

However, broader implications regarding the potential challenges faced by LRMs in real-world applications remain underexplored. A prominent issue is the problem of **overthinking** [? ? ], where excessive computational resources are expended on simple queries with negligible performance gains, even for trivial inputs such as a single word (e.g. "Hello"). These findings highlight several pressing limitations of current LRMs and raise a critical question:

---

[*] Equal contribution. ◇ Corresponding author.

39th Conference on Neural Information Processing Systems (NeurIPS 2025).

**Qwen2.5-7B-Instruct**: To find the largest possible value of $|a| + |b| + |c|$ given that $|ax^2 + bx + c| \leq 1$ for all $0 \leq x \leq 1$, we need to analyze the constraints imposed by this inequality at specific points.
...
All conditions are satisfied. Now, calculate $|a| + |b| + |c|$: $|a| + |b| + |c| = |2| + |-2| + |1| = 2 + 2 + 1 = 5$ Thus, the largest possible value of $|a| + |b| + |c|$ is $\boxed{5}$. ✗

Figure 1: Example responses from Qwen2.5-7B-Instruct, DeepSeek-R1-Distill-Qwen-7B, and our LHRMs-7B on a reasoning related task (top) and a daily QA task (bottom). While LLMs respond concisely to simple queries, they struggle with complex reasoning. LRMs handle reasoning well via explicit thinking steps but often overuse them on simple tasks, resulting in slower responses and degraded user experience. In contrast, LHRMs adaptively determines when to engage in thinking, preserving strong reasoning ability while enabling faster, more natural interactions in everyday scenarios. Additional examples are provided in Appendix G.

### *How to build a hybrid thinking system that can achieve an optimal balance between system 2 reasoning and system 1 ability?*

To fill these gaps and make LRMs more efficient, intelligent, and better aligned with human usage requirements, we draw inspiration from human cognitive processes, where complex problems often require deliberate thinking while simple ones can be handled intuitively.

Building on this insight, we introduce **L**arge **H**ybrid-**R**easoning **M**odels (LHRMs), a novel class of models that, to the best of our knowledge, are the first to explicitly address the limitations of existing LRMs by dynamically deciding whether to invoke extended thinking based on the semantic and contextual characteristics of user queries. As shown in Figure 1, this adaptive thinking mechanism allows LHRMs to balance computational efficiency with task complexity, better aligning with real-world deployment scenarios and enhancing user experience without compromising performance. To achieve this, we propose a two-stage training pipeline. The first stage, Hybrid Fine-Tuning (HFT), provides an effective cold start by enabling the model to learn to support two distinct thinking modes on the same query without mode collapse. This is followed by Hybrid Group Policy Optimization, a variant of online RL designed to teach the model when to engage in extended thinking while simultaneously generating more helpful and harmless responses. Furthermore, we introduce a metric termed Hybrid Accuracy, which correlates strongly with human expert judgment, to quantitatively evaluate the model's capability for hybrid thinking.

Extensive experimental results and human studies conducted on Qwen-2.5 series models ranging from 1.5B to 7B parameters across multiple domains (including mathematics, programming, and general tasks) demonstrate that our LHRMs effectively performs hybrid thinking by adapting to queries of varying difficulty and types. Moreover, LHRMs outperforms existing LRMs and LLMs

in both reasoning and general capabilities, while significantly improving efficiency. In general, our main contribution is as follows:

1. We introduce Large Hybrid-Reasoning Models, the first kind of model which can adaptively determine whether to perform thinking based on the contextual information of user queries, mimicking human reasoning patterns and mitigating the over-thinking problem.
2. We develop a two-stage training pipeline comprising Hybrid Fine-Tuning as a cold start and a novel RL method, Hybrid Group Policy Optimization, which enables the model to learn when to engage in thinking while simultaneously generating more helpful and harmless responses.
3. Extensive experiments across diverse model scales and domains demonstrate that LHRMs accurately performs hybrid thinking and significantly improves both efficiency and general capabilities compared to existing LRMs and LLMs.

## 2 Large Hybrid-Reasoning Models

### 2.1 Problem Formulation

Specifically, let $q$ be an input query. The LHRMs has access to two thinking modes: *Thinking* and *No-Thinking*, denoted as $\mathcal{M} = \{\vdash, \nvdash\}$, respectively. Each mode defines a conditional distribution $\mathcal{P}(a \mid q, m)$ over answers $a \in \mathcal{A}$, i.e.,

$$\mathcal{P}(a \mid q, m), \quad m \in \mathcal{M}. \tag{1}$$

For each query $q$, the model aims to select the optimal reasoning mode $m^*(q)$ from the candidate set $\mathcal{M}$, such that a task-specific utility function $\mathcal{U}(q, a)$ is maximized in expectation:

$$m^*(q) = \arg\max_{m \in \mathcal{M}} \mathbb{E}_{a \sim \mathcal{P}(a|q,m)} \Big[ \mathcal{U}(q, a) \Big]. \tag{2}$$

The overall objective is to learn a policy $\pi : \mathcal{P} \to \mathcal{M}$ to select a mode $m \in \mathcal{M}$ that maximizes the expected utility $\mathcal{U}(q, A)$ over the query distributions $\Theta = \{(\mathcal{D}_i, \mathcal{U}_i)\}_{i=1}^{N}$ :

$$\max_{\pi} \ \frac{1}{N} \sum_{i=1}^{N} \mathbb{E}_{\mathcal{D}_i \sim \Theta, \ \mathcal{D}_i \Leftrightarrow \mathcal{U}_i} \left[ \mathbb{E}_{a \sim \mathcal{P}(a|q,\pi(q)), \ q \sim \mathcal{D}_i} \Big[ \mathcal{U}_i(q, a) \Big] \right]. \tag{3}$$

We summarize two key challenges and our corresponding solutions as follows:

**C1. How to learn the policy $\pi$ based on the $\mathcal{U}_i$.** We first propose a two-stage training phase to learn an effective LHRMs by: Hybrid Fine-Tuning as cold start (§ 2.2), followed by online reinforcement learning with Hybrid Group Policy Optimization (§ 2.3) to guide LHRMs to more effectively select between two reasoning modes to generate more appropriate and high-quality responses, respectively.

**C2. How to evaluate the hybrid thinking capability of a policy $\pi$.** We propose an evaluation metric named hybrid accuracy, which is designed to assess the hybrid thinking ability of models in § 2.4.

### 2.2 Stage I: Hybrid Fine-Tuning

We begin with a hybrid-formatted supervised fine-tuning stage named Hybrid Fine-Tuning (HFT) that integrates both reasoning-intensive (*Thinking*) and direct-answer (*No-Thinking*) data. This approach mitigates the instability often observed in cold-start scenarios [15], and establishes a robust initialization for next stage reinforcement learning.

**Data Construction**. The hybrid fine-tuning dataset consists of both reasoning-intensive and direct-answer examples. The think-style subset includes high-quality math, code, and science questions sourced from existing datasets [35, 12, 39, 45, 50], with answers generated by Deepseek-R1 [15] and verified for correctness. Each instance is annotated with <think> and </think> tags to mark intermediate reasoning steps. For the non-think-style subset, we collect simple queries from WildChat-1M [58] using a FastText-based classifier [21] to exclude complex reasoning tasks. The remaining factual and conversational examples are wrapped with <no_think> and </no_think> tags to indicate that direct answers are sufficient.

After deduplication and the removal of overlaps with evaluation benchmarks, we obtain a final set of 1.7M hybrid-formatted training examples. This curated collection provides a high-quality,



**Algorithm 1** Hybrid Group Policy Optimization

**Input** model trained at Stage I $\pi_{\theta_{\text{HFT}}}$; reward models $R_\phi$; queries $\mathcal{P}$; hyperparameters $\epsilon, \beta, \mu$

1: policy model $\pi_\theta \leftarrow \pi_{\theta_{\text{HFT}}}$
2: **for** iteration = 1, ..., I **do**
3:     reference model $\pi_{\text{ref}} \leftarrow \pi_\theta$
4:     **for** step = 1, ..., M **do**
5:         Sample a batch $\mathcal{P}_b$ from $\mathcal{P}$
6:         Update the old policy model $\pi_{\theta_{\text{old}}} \leftarrow \pi_\theta$
7:         Sample $N$ outputs $\mathcal{O} \sim \pi_{\theta_{\text{old}}}(\cdot \mid q, m)$ for $q \in \mathcal{P}_b$ and $m \in \mathcal{M}$
8:         Compute rewards $r(o_i^m)$ for $o_i^m \in \mathcal{O}$ by running $R_\phi$
9:         Assign each $r(o_i^m)$ to $r_{\text{inter}}(o_i^m)$ and $r_{\text{intra}}(o_i^m)$ using Eq. 9
10:        Compute $A_i^t$ for the $t$-th token of $o_i^m$ through Eq. 10
11:        **for** iteration = 1, ..., $\mu$ **do**
12:           Update the $\pi_\theta$ by maximizing the objective $\mathcal{J}_{\text{HGPO}}$ in Eq. 11
13:        **end for**
14:     **end for**
15: **end for**
**Output** $\pi_\theta$

Figure 2: Demonstration of Hybrid Group Policy Optimization. HGPO proceeds by: (1) sampling multiple responses for each query $q$ using both reasoning modes; (2) scoring the responses with the reward model and assigning these rewards based on Eq. 9; and (3) computing the advantage and policy loss, followed by updating the policy model. AE denotes advantage estimator and reward assignment denotes Eq. 9.

format-consistent foundation for cold-start training, enabling the model to effectively handle both reasoning-heavy and direct-response tasks.

**Optimize Objective.** HFT trains the model to predict the next token based on prior context. For the constructed dataset $\mathcal{D}_{\text{HFT}} = \{(x^i, y^i)\}_{i=1}^N$, the objective is:

$$\mathcal{L}_{\text{HFT}}(\theta) = -\mathbb{E}_{[(x,y) \sim \mathcal{D}_{\text{HFT}}]} \left[ \sum_{t=1}^{|y|} \log \pi_\theta(y_t \mid x, y_{1:t-1}) \right], \tag{4}$$

### 2.3 Stage II: Hybrid Group Policy Optimization

After Stage I, LHRMs acquires an initial ability to support two distinct reasoning modes in $\mathcal{M}$ on the same query $q$ without collapsing. However, our ultimate goal is to enable the LHRMs to adaptively select the most appropriate reasoning mode $m^*(q)$ (Eq. 2) for each query $q$ to improve its applicability in real-world scenarios. To this end, we propose an effective RL algorithm named Hybrid Group Policy Optimization (HGPO) to explicitly teach the model how to adaptively select between reasoning modes, i.e., to learn the policy $\pi$ in Eq. 3, while simultaneously improving the model's foundational ability (e.g., helpfulness and harmlessness).

HGPO is illustrated in Figure 2 and Algorithm 1. Following prior RL approaches [1, 17, 41], HGPO discards the critic model and instead estimates the values using multiple samples for each prompt $q$ to reduce the computational cost training. Specifically, for each question $q$ in the task prompt set $\mathcal{P}$, HGPO samples two groups of outputs $\mathcal{O}$ from the old policy $\pi_{\theta_{\text{HFT}}}$, using *Thinking* and *No-Thinking* mode, respectively.

**Sampling Strategy**. Given a query $q \in \mathcal{P}$, we sample $N$ candidate responses from the old policy $\pi_{\theta_{\text{HFT}}}$ under two distinct reasoning modes $\mathcal{M} = \{\vdash, \nvdash\}$. Specifically,

$$\{o_i^\vdash\}_{i=1}^{N/2} \sim \pi_{\theta_{\text{HFT}}}(\cdot \mid q, m = \vdash), \quad \{o_i^\nvdash\}_{i=1}^{N/2} \sim \pi_{\theta_{\text{HFT}}}(\cdot \mid q, m = \nvdash) \tag{5}$$

We define the complete output candidate set as:

$$\mathcal{O}(q) = \{o_i^\vdash\}_{i=1}^{N/2} \cup \{o_i^\nvdash\}_{i=1}^{N/2}. \tag{6}$$

**Reward Scoring and Assignment.** A reward function $R_\phi$[2] is applied to each candidate output, yielding two reward sets:

$$\mathcal{R}^\vdash = \left\{r(o_i^\vdash)\right\}_{i=1}^{N/2}, \quad \mathcal{R}^\nvdash = \left\{r(o_i^\nvdash)\right\}_{i=1}^{N/2}. \tag{7}$$

Since the reward scores assigned by the reward model may vary significantly across different domains, we apply a rule-based assignment scheme to normalize the existing reward scores. Specifically, we compute two types of binary reward: **inter-group rewards** $r_{\text{inter}}$ and **intra-group rewards** $r_{\text{intra}}$ to jointly capture both the relative quality across different reasoning modes and the answer quality within each individual reasoning mode. Given the average reward for each mode computed as

$$\bar{\mathcal{R}}^\vdash = \frac{2}{N}\sum_{i=1}^{N/2} r(o_i^\vdash), \quad \bar{\mathcal{R}}^\nvdash = \frac{2}{N}\sum_{i=1}^{N/2} r(o_i^\nvdash), \tag{8}$$

we have $r_{\text{inter}}$ and $r_{\text{intra}}$ assigned as:

$$r_{\text{inter}}(o_i^m) = \begin{cases} 1, & \text{if } m = \underset{m'\in\{\vdash,\nvdash\}}{\arg\max}\{\bar{\mathcal{R}}^\vdash, \bar{\mathcal{R}}^\nvdash + \delta\} \\ 0, & \text{otherwise} \end{cases}, \quad r_{\text{intra}}(o_i^m) = \begin{cases} 1, & \text{if } i = \underset{j\in\{1,...,N/2\}}{\arg\max} r_j^m \\ 0, & \text{otherwise} \end{cases}. \tag{9}$$

Here margin $\delta$ is a hyperparameter that governs the trade-off between two reasoning modes.

**Advantage Estimation.** After getting the final two kinds of rewards for $\mathcal{O}$, we can adopt advantage estimators like REINFORCE++ [17], GRPO [41], and RLOO [1] to estimate the advantages. Here we use GRPO as the default advantage estimator, i.e.,

$$A_i^t = \underbrace{\left[\frac{r_{\text{intra}}(o_i) - \text{mean}(r_{\text{intra}}(o_j))}{\text{std}(r_{\text{intra}}(o_j))}\right]}_{\text{GRPO for intra-group advantage } A_{\text{intra}}} + \mathbb{1}\{o_i^t \in \Phi\} \cdot \alpha \underbrace{\left[\frac{r_{\text{inter}}(o_i) - \text{mean}(r_{\text{inter}}(o_j))}{\text{std}(r_{\text{inter}}(o_j))}\right]}_{\text{GRPO for inter-group advantage } A_{\text{inter}}}, \tag{10}$$

where $A_i^t$ is the final estimated per-token advantage for each response in $\mathcal{O}$, $\Phi = \{\texttt{<think>}, \texttt{<no\_think>}\}$ and $\alpha$ is the hyperparameter for controlling the weight of inter-group advantages. We also explore the impact of using different estimators in § 3.3.

**Optimize Objective.** Following [1, 17, 41], HGPO optimizes the policy model $\pi_\theta$ by maximizing the following objective:

$$\mathcal{J}_{\text{HGPO}}(\theta) = \mathbb{E}_{\left[q\sim\mathcal{P}, \{o_i^m\}_{i=1}^N \sim \pi_{\theta_{\text{HFT}}}(\mathcal{O}|q), m\in\mathcal{M}\right]}$$

$$\frac{1}{N}\sum_{i=1}^{N}\sum_{t=1}^{|o_i|}\left[\min\left(\frac{\pi_\theta(o_i^{m,t}|q, o_i^{m,<t})}{\pi_{\theta_{\text{HFT}}}(o_i^{m,t}|q, o_i^{m,<t})}A_i^t, \text{clip}\left(\frac{\pi_\theta(o_i^{m,t}|q, o_i^{m,<t})}{\pi_{\theta_{\text{HFT}}}(o_i^{m,t}|q, o_i^{m,<t})}, 1-\epsilon, 1+\epsilon\right)A_i^t\right) - \beta\mathbb{D}_{\text{KL}}(\pi_\theta||\pi_{\text{ref}})\right], \tag{11}$$

where

$$\mathbb{D}_{\text{KL}}(\pi_\theta||\pi_{\text{ref}}) = \frac{\pi_{\text{ref}}(o_i^m|q)}{\pi_\theta(o_i^m|q)} - \log\frac{\pi_{\text{ref}}(o_i^m|q)}{\pi_\theta(o_i^m|q)} - 1. \tag{12}$$

Here $\pi_{\theta_{\text{HFT}}}$ denotes the model we get from Stage I in § 2.2 and $\epsilon$ and $\beta$ are hyper-parameters.

## 2.4 Evaluating Hybrid Thinking Capability

To more comprehensively evaluate the performance of LHRMs, beyond conventional downstream task metrics, we propose a new metric called **Hybrid Accuracy** ($\mathcal{H}_{\text{Acc}}$), which aims to measure the LHRMs' ability to correctly select the appropriate reasoning pattern.

Given the task prompt set $\mathcal{P} = \{p_i\}_{i=1}^K$, LHRMs is first applied to sample $N$ responses under two distinct reasoning modes $\vdash$ and $\nvdash$. A reward model is then employed as a scorer to evaluate and assign scores to both sets of generated responses. The mode with the higher average score is regarded as the ground-truth preferred reasoning mode for each prompt, denoted as $m_{\text{gt}}$. In cases where the average scores of the two modes are equal or the difference between them falls below a predefined margin, the mode yielding the shorter response is selected as $m_{\text{gt}}$. Subsequently, we allow LHRMs to autonomously select a reasoning mode $m_{\text{p}}$ for each prompt. The proportion of prompts for which LHRMs' selected mode matches the ground-truth mode is reported as

$$\mathcal{H}_{\text{acc}} = \frac{1}{K}\sum_{i=1}^{K}\mathbb{1}\left[\text{Equal}(m_{\text{gt}}, m_{\text{p}})\right] \quad \text{s.t.} \quad m_{\text{gt}}, m_{\text{p}} \in \{\vdash, \nvdash\}. \tag{13}$$

---

[2]For queries with definitive answers, we use rule-based rewards [41, 15] for a better reward estimation; otherwise, a trained parametric reward model is applied.

# 3 Experimental Results

## 3.1 Experimental Setup

**Compared Baselines.** To validate the effectiveness of our proposed method LHRMs, we conduct a comprehensive comparison against state-of-the-art LLMs and LRMs derived from the same base models. Specifically, we build our LHRMs on both 1.5B and 7B parameter versions of Qwen-2.5-math-base [52], and compare our method with several Qwen-2.5-derived variants, including:

- **LLMs**: we compare our model with Qwen-2.5 Math series [52] and Instruct series [51], which show great capabilities in coding, mathematics and general tasks.
- **LRMs**: Here we use DeepSeek-R1-Distill series [10], which are distilled using the reasoning data generated by DeepSeek-R1[10] and attain strong reasoning ability.
- **Hybrid**: Due to the absence of existing models capable of hybrid thinking, we compare our final model against the baseline we obtained in our Stage I (§ 2.2) (denoted as HFT). Additionally, we construct two further baselines by applying Direct Preference Optimization (DPO) [40] and Rejection Sampling Fine-Tuning (RFT) [55] to the checkpoint $\pi_{\theta_{HFT}}$ from Stage I, using the same training data as in Stage II (§ 2.3). These baselines are referred to as HFT-DPO and HFT-RFT, respectively. Implementation details of DPO and RFT can be found in Appendix A.

**Evaluation Settings.** We primarily evaluate model performance based on the following aspects:

- **Reasoning Capabilities**. We evaluate models on a comprehensive set of widely-used reasoning benchmarks, including math related like MATH500 [26], AIME24 [2], AMC23, Olympaid Bench [16], and code related like LiveCodeBench [20], MBPP [4] and MBPP+ [30].
- **General Capabilities**. We assess the models' general capabilities through open-ended generation tasks using LLMs as judges. Specifically, we adopt AlpacaEval 2.0 [11] and Arena-Hard [25] to assess instruction-following ability and alignment with human preferences.
- **Hybrid Thinking Capabilities**. We compute Hybrid Accuracy ($\mathcal{H}_{Acc}$) (§ 2.4) on MATH500 to evaluate the model's ability to correctly select the appropriate reasoning mode.

More details about the evaluation settings can be found in Appendix D.

**Training Settings.** For stage I, we finally obtained 1.7 M SFT data and details about construction pipeline, sources and statistics can be found in § 2.2 and Appendix C. All models are trained for 3 epochs with the AdamW optimizer, employing a 10% linear warmup followed by a cosine learning rate decay schedule. The maximum learning rate is set to 1e−4, with a batch size of 128 and a maximum sequence length of 32k tokens. Training the 7B model in the SFT phase takes approximately 2.5 days on 4 nodes of NVIDIA 8×H100 stations.

For stage II, we construct the training dataset by randomly sampling 76K queries from Deepscaler [32] and Tülu3 [23] (details can be found in Appendix C). We use Llama-3.1-Tulu-3-8B-RM [3] as the parametric reward model in Eq. 7. We use VeRL [43] to conduct experiments. By default, we use a constant $1 \times 10^{-6}$ learning rate together with AdamW optimizer for policy model, and use a batch size of 256 and micro batchsize of 8. The rollout stage collects 256 prompts and samples 4 responses for each prompt. We set $\alpha = 1.0$ and margin $= 0.2$ for RL training. We set KL coefficient to 0.001 and $\epsilon = 0.5$ in Eq. 11 in all experiments. The RL phase takes 2 days on NVIDIA 4×H100 Stations.

## 3.2 Main Results

**Overall Performance.** The overall results of different models on the aforementioned benchmarks are presented in Table 1. We observe that LHRMs consistently outperform all comparable baselines across both the 1.5B and 7B model scales, achieving average improvements of **9.2%** and **7.1%** compared to HFT-DPO, and **18.3%** and **11.5%** compared to HFT-RFT, at the 1.5B and 7B scales, respectively. Specifically, it surpasses the strongest competing baseline, HFT-DPO, by **10.3%** and **13.6%** on the AIME24 benchmark at 1.5B and 7B scales, respectively. On the 7B scale, LHRMs further outperforms HFT-DPO by **50.2%** on Alpaca and **93.4%** on Arena-Hard. Notably, AIME24 and Arena-Hard represent the most challenging benchmarks in the math and general domains, respectively. These results demonstrate the strong reasoning and general capabilities of our LHRMs.

---

[3]https://huggingface.co/allenai/Llama-3.1-Tulu-3-8B-RM

Table 1: Performance comparison across different tasks. "-" indicates that the model does not support hybrid thinking and is therefore left blank. The last column (Avg.) reports the average performance across all evaluation tasks, excluding the $\mathcal{H}_{acc}$ metric. Bold numbers indicate the best performance. *Type* refers to the model's reasoning mode, where *Hybrid* denotes models that adaptively select between *Thinking* and *No-Thinking* modes for each query.

| Methods | Type | MATH | | | | Code | | | General | | $\mathcal{H}_{acc}$ | Avg. |
| | | MATH500 | AIME24 | AMC23 | Olympiad | LiveCode | MBPP | MBPP+ | Alpaca | Arena | | |
|---|---|---|---|---|---|---|---|---|---|---|---|---|
| **1.5B size model** | | | | | | | | | | | | |
| Qwen2.5-Math-1.5B | LLMs | 42.4 | 3.3 | 22.5 | 16.7 | 0.4 | 16.1 | 14.3 | 0.1 | 1.8 | - | 13.1 |
| Qwen2.5-1.5B-Instruct | LLMs | 51.0 | 3.3 | 52.8 | 38.7 | 2.2 | 60.1 | 51.9 | 8.8 | 1.1 | - | 30.0 |
| Qwen2.5-Math-1.5B-Instruct | LLMs | 72.0 | 6.7 | 60.0 | 38.1 | 3.7 | 26.7 | 23.8 | 2.8 | 4.7 | - | 26.5 |
| DeepSeek-R1-Distill-Qwen-1.5B | LRMs | 83.9 | 28.9 | 62.9 | 43.3 | 16.8 | 54.2 | 46.3 | 5.6 | 2.7 | - | 38.3 |
| HFT-1.5B | Hybrid | **87.8** | 32.7 | **75.0** | 48.9 | 15.7 | 54.8 | 47.4 | 13.1 | 6.9 | 41.4 | 42.5 |
| HFT-RFT-1.5B | Hybrid | 82.2 | 22.0 | 67.5 | 44.1 | 14.2 | 49.7 | 42.6 | 13.6 | 8.5 | 48.1 | 38.3 |
| HFT-DPO-1.5B | Hybrid | 86.8 | 32.6 | **75.0** | 48.7 | **17.2** | 50.5 | 42.6 | 13.3 | 6.9 | 45.8 | 41.5 |
| LHRMs-1.5B | Hybrid | **87.8** | 35.3 | **75.0** | 50.4 | **17.2** | 61.1 | 54.0 | 16.9 | 10.4 | **54.4** | **45.3** |
| **7B size model** | | | | | | | | | | | | |
| Qwen2.5-Math-7B | LLMs | 57.0 | 13.3 | 22.5 | 21.8 | 6.0 | 31.5 | 27.3 | 2.0 | 7.0 | - | 20.9 |
| Qwen2.5-7B-Instruct | LLMs | 77.0 | 13.3 | 52.8 | 29.1 | 14.6 | 79.9 | 67.5 | **36.2** | 25.8 | - | 44.0 |
| Qwen2.5-Math-7B-Instruct | LLMs | 82.4 | 10.0 | 62.5 | 41.6 | 2.6 | 40.2 | 34.7 | 3.8 | 10.0 | - | 32.0 |
| DeepSeek-R1-Distill-Qwen-7B | LRMs | 92.8 | 55.5 | 91.5 | 58.1 | 37.6 | 74.3 | 64.3 | 19.1 | 17.9 | - | 56.8 |
| HFT-7B | Hybrid | 93.6 | 56.7 | **95.0** | 58.5 | 34.7 | 70.6 | 59.8 | 23.7 | 14.0 | 34.2 | 56.4 |
| HFT-RFT-7B | Hybrid | 87.8 | 55.3 | 82.5 | 55.0 | 35.8 | 81.0 | 68.8 | 28.1 | 14.0 | 49.7 | 56.6 |
| HFT-DPO-7B | Hybrid | **93.8** | 58.7 | 92.5 | 60.6 | **38.8** | 80.1 | 68.3 | 23.3 | 13.0 | 37.1 | 58.9 |
| LHRMs-7B | Hybrid | **93.8** | 66.7 | **95.0** | 61.2 | **38.8** | 81.5 | 69.6 | 35.0 | 26.0 | **71.9** | **63.1** |

Furthermore, we find that our LHRMs achieves the best hybrid thinking performance, as measured by $\mathcal{H}_{acc}$, significantly outperforming all baselines. For example, it exceeds HFT-DPO by **93.8%** and RFT by **44.7%**. This further demonstrates that our HGPO effectively enables the model to learn correct hybrid thinking behaviors, providing a promising and practical pathway for building hybrid thinking systems. Next, we provide a detailed investigation into each of the two training stages.

**Effect of HFT at Stage I**. By comparing HFT with the Qwen2.5 and Deepseek-R1-Distill series, we observe that HFT significantly enhances both reasoning and general capabilities, while demonstrating robust cold-start performance by maintaining stability during hybrid reasoning without failure or collapse. These results validate the effectiveness of the proposed HFT (§ 2.2) framework.

**Effect of HGPO at Stage II**. By comparing LHRMs with HFT, we find that HGPO further substantially improves the model's reasoning and general capabilities, while enabling more effective selection of thinking modes (over **31.4%** and **110.2%** $\mathcal{H}_{acc}$ improvements at 1.5B and 7B size, respectively). These findings demonstrate the effectiveness of HGPO.

Furthermore, when comparing LHRMs with HFT-DPO and HFT-RFT, we observe that LHRMs achieves superior downstream performance on both reasoning and general tasks, along with higher accuracy in reasoning mode selection. This highlights the effectiveness of HGPO.

Notably, we also observe that LHRMs exhibits stronger cross-domain generalization capabilities. Although RL training is conducted solely on math and general-domain data, LHRMs-1.5B achieves substantial improvements on code-related tasks, with gains of **11.1%** on MBPP and **13.9%** on MBPP+. In contrast, both HFT-DPO-1.5B and HFT-RFT-1.5B show performance drops (**7.8%** and **9.3%**, respectively) on the same tasks. This indicates that LHRMs is able to generalize learned hybrid thinking patterns across domains—a property not observed in DPO or RFT.

### 3.3 Analysis

**Effect of Different Advantage Estimators**. To investigate the impact of different advantage estimators on HGPO training, we replace GRPO in Eq. 10 with RLOO [1] and REINFORCE++ [17]. Implementation details can be found in Appendix B. As shown in Figure 3 (a), all estimators yield competitive performance, indicating that HGPO is robust to the choice of advantage estimator.

**Effect of Margin $\delta$ in Eq. 9**. We investigate how varying the margin $\delta$ influences the model's bias toward the two reasoning modes. As shown in Figure 3 (b), different values of $\delta$ result in distinct

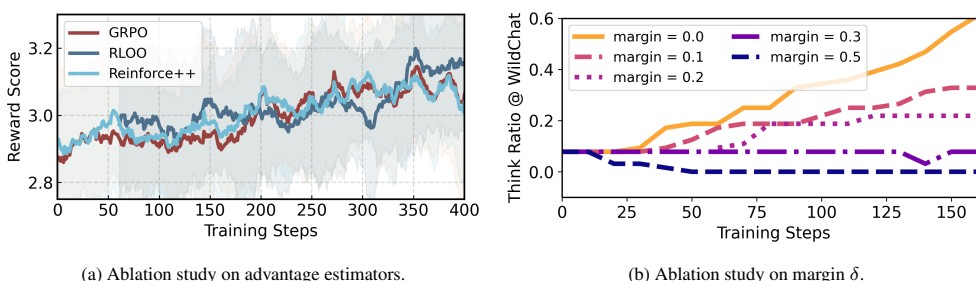

(a) Ablation study on advantage estimators.   (b) Ablation study on margin $\delta$.

Figure 3: Ablation study on the effects of advantage estimators and margin $\delta$. $\delta$ in Eq. 9.

hybrid reasoning behaviors. Specifically, a larger $\delta$ encourages the model to favor the *No-Thinking* mode. This suggests that $\delta$ can serve as a control knob for tailoring hybrid reasoning behavior to specific application needs. For instance, a higher $\delta$ may be preferred when real-time responsiveness is prioritized, whereas a lower $\delta$ is more suitable when reasoning quality is the primary concern.

**Thinking Ratio Study within Domain**. Figure 4 illustrates the distribution of thinking ratio across varying difficulty levels in the MATH500 benchmark for both HFT-7B and LHRMs-7B. We observe that HFT-7B maintains a consistently high thinking ratio across all difficulty levels. In contrast, after applying the Stage II HGPO, LHRMs-7B exhibits a decreasing thinking ratio as the difficulty level decreases while getting a higher overall performance (as shown in Table 1). This trend suggests that our HGPO can effectively enable the model to adaptively perform hybrid thinking based on the input query. Specifically, the model

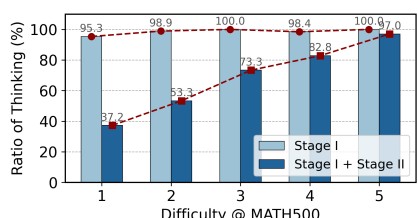

Figure 4: Analysis of thinking ratio of LHRMs within a single domain.

learns to respond to simpler queries using a *No-Thinking* strategy, thereby reducing reasoning cost and inference latency without sacrificing accuracy. For more challenging queries, the model consistently engages in full thinking, achieving higher reasoning accuracy where it matters most.

**Thinking Ratio Study across Domain**. We investigate how HGPO affects model performance on unseen domains not encountered during the RL phase. Specifically, we conduct RL using data from the math and general domains, and then evaluate the resulting models on the code domain. As shown in Figure 5, we observe that the model attained from HFT exhibits a relatively low think ratio in the code domain. However, as RL progresses—despite being conducted on unrelated domains such as math—the think ratio in the code domain gradually increases, accompanied by

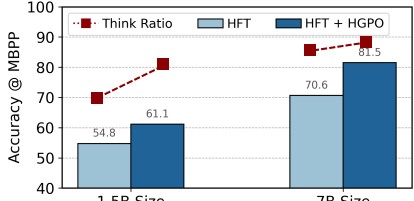

Figure 5: Analysis of thinking ratio of LHRMs across different domains.

improved performance. This suggests that our HGPO is capable of transferring hybrid thinking patterns learned in a specific domain (e.g., math) to other domains (e.g., code), demonstrating strong generalization and transferability.

**Thinking Ratio Study across Model Size**. We analyze how model size influences hybrid thinking behavior during the RL phase. In Figure 6, we present the think ratio of the 1.5B and 7B models on the MATH500 benchmark. We observe that, as RL training progresses, the think ratio of the 7B model gradually decreases, while that of the 1.5B model increases. This indicates that hybrid thinking behavior during RL is strongly correlated with model capacity and knowledge storage. Specifically, larger models with

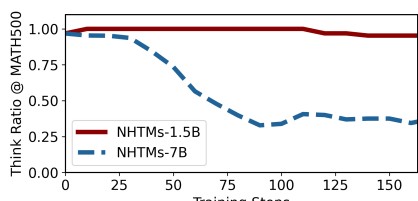

Figure 6: Ablation study on model size.

stronger knowledge reserves tend to shift towards using *No-Thinking* strategies on solvable problems to reduce reasoning overhead and improve response efficiency. In contrast, smaller models are more inclined to invoke *Thinking* more frequently, as a way to compensate for their limited capabilities through enhanced reasoning.

> **Question: If** $a \star b = 9a + 2b - ab + 5$, **what is the value of** $5 \star 1$?
>
> **DeepSeek-R1-Distill-Qwen-7B**: `<think>` Okay, let's tackle this problem. So, we have this operation defined as a star b equals 9a plus 2b minus ab plus 5. And we need to find the value of 5 star 1. Hmm, let me make sure I understand the question... (2941 characters) `</think>`
> To solve $5 \star 1$ using the defined operation $a \star b = 9a + 2b - ab + 5$:
> 1. Substitute $a = 5$ and $b = 1$ into the expression: $5 \star 1 = 9(5) + 2(1) - (5)(1) + 5$
> 2. Calculate each term: - 9 × 5 = 45 - 2 × 1 = 2 - 5 × 1 = 5
> 3. Substitute these values back into the expression: $45 + 2 - 5 + 5$
> 4. Perform the arithmetic operations step-by-step: - 45 + 2 = 47 - 47 − 5 = 42 - 42 + 5 = 47
>
> Thus, the value of $5 \star 1$ is $\boxed{47}$ ✓
>
> **LHRMs-7B**: `<no_think>` To solve the problem, we need to substitute $a = 5$ and $b = 1$ into the given function $a \star b = 9a + 2b - ab + 5$.
> Step-by-step solution:
> 1. Start with the given function: $a \star b = 9a + 2b - ab + 5$
> 2. Substitute $a = 5$ and $b = 1$ into the function: $5 \star 1 = 9(5) + 2(1) - 5(1) + 5$
> 3. Perform the multiplication operations: $5 \star 1 = 45 + 2 - 5 + 5$
> 4. Simplify the expression by performing the addition and subtraction operations: $5 \star 1 = 45 + 2 - 5 + 5 = 47$
>
> Therefore, the final answer is: $\boxed{47}$. ✓ `</no_think>`

Figure 7: Example outputs from DeepSeek-R1-Distill-Qwen-7B and our LHRMs-7B model. For easy questions, LHRMs-7B selects the *No-Thinking* mode.

**Case Analysis**. We present representative model responses in Figure 1 and 7. We observe that for relatively simple queries—such as everyday questions and basic math questions, LHRMs tends to autonomously select the *No-Thinking* mode, achieving fast and accurate responses. In contrast, for more complex problems that require deeper reasoning, LHRMs adaptively switches to the *Thinking* mode to produce more precise solutions. More example responses can be found in Appendix G.

**Additional Analysis**. To further verify the effectiveness of our method, we conduct additional analyses in Appendix E, including the impact of hybrid training on reasoning ability (E.1), an efficiency analysis (E.2), an ablation comparing HGPO with standard GRPO (E.3), and an annotation consistency validation for Hybrid Accuracy (E.4).

# 4 Related Works

**Test-Time Scaling (TTS).** Test-time scaling [12, 15, 37, 19] has been validated as a promising approach to enhance model performance beyond scaling model size [44]. There are two primary approaches to implementing TTS [37]: **Parallel** and **Sequential**. The former is achieved by sampling multiple solutions and select the one by a selector (e.g., an outcome reward model), like Best-of-N [5, 19, 24], and Monte-Carlo Tree Search (MCTS) [28, 56, 61, 8]. The latter aims to achieve TTS by enabling the model to generate longer outputs, i.e., Chain-of-Thoughts (CoT) within a single sampling pass by prompting, finetuning or reinforcement learning. Beyond the field of NLP, TTS has also been shown to effectively improve the test-time performance of trained models in other domains, like image generation, video generations [27] and multi-modality learning [13, 18, 6].

**Large Reasoning Models.** Recent advances in Large Reasoning Models (LRMs), such as DeepSeek-R1 [10], OpenAI o1/o3 series [38], and others [46, 37, 3, 14], have led to a growing focus on Large Reasoning Models (LRMs). Compared to general LLMs, LRMs extend the capabilities of LLMs by generating long chains of reasoning steps with reflection before outputting the final answers. LRMs are typically developed by applying reinforcement learning such as GRPO [15], REINFORCE++ [17], and related approaches [7, 53], or distilled from stronger models [45, 35, 12, 57, 54].

# 5 Conclusion

In this work, we focus on building a large language model that effectively balances reasoning capabilities and general-purpose performance. To this end, we propose a novel evaluation metric, $\mathcal{H}_{acc}$, designed to consistently assess a model's ability to perform hybrid thinking across diverse tasks. We further introduce a two-stage training pipeline consisting of HFT and HGPO. Experimental results demonstrate that this pipeline significantly improves performance on both reasoning-centric and general downstream tasks. Moreover, it enhances the model's hybrid thinking capabilities, leading to a better user experience by reducing unnecessary reasoning on simple queries commonly observed in LRMs, and mitigating the insufficient reasoning capabilities in conventional LLMs.

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

# Contents

# A  Implementation Details for DPO and RFT

In this section, we detail the construction pipeline of training data for both DPO and RFT used in § 3. To construct datasets for DPO and RFT, we adopt the same set of queries used in HGPO for offline sampling. Specifically, for each query $q$, we sample two responses using each of the two thinking modes $m \in \{\vdash, \nvdash\}$, resulting in a set of four responses per query as

$$\mathcal{O}(q) = \left\{ o_i^{\vdash} \right\}_{i=1}^2 \cup \left\{ o_i^{\nvdash} \right\}_{i=1}^2. \tag{14}$$

Then the reward function $R_\phi$[4] in Eq. 15 is applied to score these responses as:

$$\mathcal{R}^{\vdash} = \left\{ r(o_i^{\vdash}) \right\}_{i=1}^2, \quad \mathcal{R}^{\nvdash} = \left\{ r(o_i^{\nvdash}) \right\}_{i=1}^2. \tag{15}$$

Given the average reward for each mode computed as

$$\bar{\mathcal{R}}^{\vdash} = \frac{1}{2} \sum_{i=1}^2 r(o_i^{\vdash}), \quad \bar{\mathcal{R}}^{\nvdash} = \frac{1}{2} \sum_{i=1}^2 r(o_i^{\nvdash}), \tag{16}$$

**DPO**. For DPO, the training dataset which contains win and lose sample is constructed as following:

$$\mathcal{D}_{\text{DPO}} = \left\{ (q, o_w, o_l) \,\middle|\, o_w = \underset{o \in \{o_i^{m_w}\}_{i=1}^2}{\arg\max} \left( r\left(o\right) \right), \quad o_l = \underset{o \in \{o_i^{m_l}\}_{i=1}^2}{\arg\min} \left( r\left(o\right) \right) \right\}, \tag{17}$$

while

$$m_w = \underset{m \in \{\vdash, \nvdash\}}{\arg\max} \bar{\mathcal{R}}^m, \quad m_l = \underset{m \in \{\vdash, \nvdash\}}{\arg\min} \bar{\mathcal{R}}^m. \tag{18}$$

After getting $\mathcal{D}_{\text{DPO}}$, we optimize the model $\pi_\theta$ initialized from $\pi_{\theta_{\text{HFT}}}$ by using training objective:

$$\max_{\pi_\theta} \mathbb{E}_{(x, o_w, o_l) \sim \mathcal{D}_{\text{DPO}}} \left[ \log \sigma \left( \beta \log \frac{\pi_\theta(o_w \mid x)}{\pi_{\theta_{\text{HFT}}}(o_w \mid x)} - \beta \log \frac{\pi_\theta(o_l \mid x)}{\pi_{\theta_{\text{HFT}}}(o_l \mid x)} \right) \right]. \tag{19}$$

**RFT**. For RFT, the training dataset is constructed as following:

$$\mathcal{D}_{\text{RFT}} = \left\{ (q, o) \,\middle|\, o = \underset{o \in \mathcal{O}}{\arg\max} \left( \mathcal{R}^{\vdash} \cup \mathcal{R}^{\nvdash} \right) \right\}. \tag{20}$$

The training objectives for RFT are presented as:

$$\mathcal{L}_{\text{RFT}}(\theta) = -\mathbb{E}_{[(x, o) \sim \mathcal{D}_{\text{RFT}}]} \left[ \sum_{t=1}^{|o|} \log \pi_\theta(o_t \mid x, o_{1:t-1}) \right], \tag{21}$$

For implementations, we use LLaMA-Factory [60] [5] as the codebase for both DPO and RFT. To ensure a fair comparison, we maintain the same learning rate, batch size, and total number of training samples as used in Stage II of HGPO.

# B  Implementation Details for RLOO and Reinforce++

First of all, We compare different advantage estimators including REINFORCE++ [17], GRPO [41], and RLOO [1], toggling the existence of our HGPO. To make different algorithms compatible with the compound of intra-group rewards and inter-group rewards, we accordingly make adaptions similar to Eq. 10. For Reinforce++, we have

$$A_i^t = \underbrace{\sum_{s=t}^{|o_i|} \gamma^{s-t} \cdot r_{\text{intra}}\left(o_i\right)}_{\text{REINFORCE++ for intra-group advantage } A_{\text{intra}}} + \underbrace{\mathbb{1}\{o_i^t \in \Phi\} \cdot \alpha \left[ r_{\text{inter}}\left(o_i\right) \right]}_{\text{REINFORCE++ for inter-group advantage } A_{\text{inter}}}, \tag{22}$$

---

[4]Same with § 2.3, for queries with definitive answers, we use rule-based rewards [41, 15]; otherwise, a trained parametric reward model [23] is applied.

[5]https://github.com/hiyouga/LLaMA-Factory.git

Here is a hyperparameter representing the decay factor, which is set to 0.99 in our experiments. For RLOO, we have

$$A_i^t = \underbrace{\left[ r_{\text{intra}}\left(o_i\right) - \frac{1}{N-1}\sum_{j\neq i} r_{\text{intra}}\left(o_j\right) \right]}_{\text{RLOO for intra-group advantage } A_{\text{intra}}} + \underbrace{\mathbb{1}\{o_i^t \in \Phi\} \cdot \alpha \left[ r_{\text{inter}}(o_i) - \frac{1}{N-1}\sum_{j\neq i}\left(r_{\text{inter}}\left(o_j\right)\right) \right]}_{\text{RLOO for inter-group advantage } A_{\text{inter}}}. \quad (23)$$

From § 3.3, we show that **HGPO is a general plug-in for almost any advantage estimator**, which largely extends the use cases of HGPO. We implement both RLOO and Reinforce++ on VeRL [43] [6].

## C  Dataset Statistics

**Stage I**. Figure 8 shows the token length distributions for the *Thinking* and *No-Thinking* datasets we construct in Stgae I. The *Thinking* data has an average length of 575 tokens, with the 25th and 75th percentiles at 362 and 672 tokens, respectively, and a maximum length of 9,148 tokens. In contrast, the *No-Thinking* data exhibits a significantly higher average length of 4,897 tokens, with the 25th and 75th percentiles at 1,719 and 6,738 tokens, and a maximum of 23,997 tokens.

We present the scores and statistics of our constructed dataset for HFT in Table 2. The dataset covers a diverse range of domains, primarily including reasoning-intensive tasks such as mathematics and code, as well as general-purpose question answering.

**Stage II**. We report the details of the training data used in Stage II in Table 3.

Table 2: Data distribution and source of Stage I.

| Category | Source | Data Size | Total | Link |
|---|---|---|---|---|
| General | WildChat-1M [58] | 649,569 | 674,908 | https://huggingface.co/datasets/allenai/WildChat-1M |
|  | OSSAT2 [22] | 25,339 |  | https://huggingface.co/datasets/OpenAssistant/oasst1 |
| MATH | SYNTHETIC-1 [35] | 343,988 | 631,325 | https://huggingface.co/datasets/PrimeIntellect/SYNTHETIC-1 |
|  | OpenR1-Math [12] | 93,533 |  | https://huggingface.co/datasets/open-r1/OpenR1-Math-220k |
|  | OpenThought [45] | 55,566 |  | https://huggingface.co/datasets/open-thoughts/OpenThoughts-114k |
|  | AoPS [34] | 137,497 |  | https://huggingface.co/datasets/di-zhang-fdu/AOPS |
|  | AIME | 741 |  | https://huggingface.co/datasets/di-zhang-fdu/AIME_1983_2024 |
| Coding | SYNTHETIC-1 [35] | 107,543 | 381,845 | https://huggingface.co/datasets/PrimeIntellect/SYNTHETIC-1 |
|  | OpenR1-Codeforces [39] | 8,926 |  | https://huggingface.co/datasets/open-r1/codeforces |
|  | OpenThought [45] | 19,447 |  | https://huggingface.co/datasets/open-thoughts/OpenThoughts-114k |
|  | KodCode [50] | 245,929 |  | https://huggingface.co/datasets/KodCode/KodCode-V1 |
| Others | SYNTHETIC-1 [35] | 6,508 | 6,508 | https://huggingface.co/datasets/PrimeIntellect/SYNTHETIC-1 |
| Total | – | 1,694,586 | 1,694,586 | – |

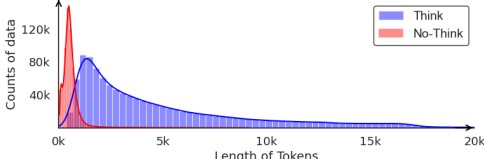

Figure 8: Token length distributions of *Thinking* and *No-Thinking* data in Stage I.

Table 3: Data distribution and source of Stage II. Note that our experiments are conducted on subsets of the two datasets rather than their full versions.

| Category | Source | Data Size | Used |
|---|---|---|---|
| General | Tülu 3 [23] Tülu 3 preference dataset | 337,000 | 50,000 |
| MATH | DeepScaler [32] DeepScaleR-Preview-Dataset | 40,300 | 40,000 |

## D  Evaluation Settings

In this section, we introduce the details for downstream tasks evaluation in § 3. During inference, we set the temperature to 0.6, the top-p value to 0.95, and the maximum generation length to 35,000 tokens.

---

[6] https://github.com/volcengine/verl.git

- For MATH500, AIME24, GPQA Diamond, and LiveCodeBench, we adopt the OpenR1 [12] evaluation framework. For MBPP and MBPP+, we use the EvalPlus [29, 30] framework with Chain-of-Thought (CoT) [47] prompting. We report pass@1 for all reasoning related benchmarks.
- For AlpacaEval 2.0, we use GPT-4-Turbo as the baseline model, while for Arena-Hard, we adopt evaluation queries from Arena-Hard-v2.0 and use GPT-4o-mini as the baseline. To reduce length-related bias, only the final generated summary is submitted to the judge in think mode.

We provide a summarization and links of evaluation tools we used in experiments in Table 4.

Table 4: Summarization of evaluation tools we used in experiments.

| Task | Tool Name | Source |
|---|---|---|
| MATH500 [26], AIME24 [2], Livecodebench [20] | open-r1 [12] | https://github.com/huggingface/open-r1 |
| Olympaid Bench, AMC23 | math-evaluation-harness | https://github.com/ZubinGou/math-evaluation-harness |
| MBPP [4, 30], MBPP+ [4, 30] | evalplus [29, 30] | https://github.com/evalplus/evalplus |
| AlpacaEval [11] | aplaca_eval | https://github.com/tatsu-lab/alpaca_eval |
| ArenaHard v2.0 [25] | arena-hard-auto | https://github.com/lmarena/arena-hard-auto |

# E   Additional Experiments

## E.1   Comparison between Hybrid and Non-Hybrid Models

To further validate the robustness of our hybrid training framework, we perform a controlled comparison between hybrid-trained and non-hybrid models to assess whether hybrid training compromises reasoning capability.

Specifically, we fine-tuned both the 1.5B and 7B variants of *Qwen-2.5-Math* on approximately 1.1M math and code reasoning samples formatted in the *Thinking-style*. These models were trained using identical optimization settings as our hybrid models, allowing for a direct and fair comparison. The two model configurations are defined as follows:

**PureThink-SFT models.**   These models are trained purely on ∼1.1M *Thinking-style* math and code data using standard supervised fine-tuning (SFT), without any hybrid formatting or No-Thinking examples.

**HFT (Think-Only) models.**   These models are trained using our Hybrid Fine-Tuning (HFT) pipeline, which jointly incorporates both Thinking and No-Thinking data. During inference, the models are constrained to always operate in the Thinking mode to enable a direct comparison with PureThink-SFT.

As presented in Table 5, both model types achieve nearly identical results across multiple reasoning benchmarks at both the 1.5B and 7B scales. This consistency indicates that hybrid training does not impair reasoning performance. Instead, it provides the additional flexibility of supporting adaptive reasoning modes without compromising capability.

Table 5: Comparison between Hybrid and Non-Hybrid Models on Reasoning-Intensive Benchmarks.

| Model (Training Type) | MATH500 | AIME24 | AMC23 | Olympiad | LiveCodeBench | MBPP | MBPP+ | Avg. |
|---|---|---|---|---|---|---|---|---|
| PureThink-SFT-1.5B | 87.6 | 34.7 | 72.5 | 49.9 | 17.2 | 63.5 | 53.2 | 54.1 |
| HFT-1.5B (Think Only) | 87.8 | 32.7 | 75.0 | 50.0 | 16.8 | 62.2 | 52.9 | 53.9 |
| PureThink-SFT-7B | 94.0 | 56.7 | 92.5 | 59.9 | 38.7 | 80.7 | 68.3 | 70.1 |
| HFT-7B (Think Only) | 93.6 | 56.7 | 95.0 | 59.7 | 38.4 | 80.3 | 68.9 | 70.4 |

## E.2   Efficiency and Compression Analysis

To further analyze the efficiency of the proposed hybrid reasoning approach, we compared the average *thinking token length* and corresponding accuracies of our models against several leading Chain-of-Thought (CoT) compression methods, including **O1-Pruner** [31], **CoT-Valve** [33], and **DAST** [42].

As shown in Table 6, our Large Hybrid-Reasoning Models (LHRMs) achieve a better balance between compression and accuracy. Compared to full-length thinking baselines (`HFT-1.5B-Think` and `HFT-7B-Think`), LHRMs significantly reduce reasoning lengths across most tasks while maintaining or improving performance.

Although some prior CoT compression approaches produce shorter outputs (e.g., O1-Pruner, DAST), they often exhibit accuracy degradation. In contrast, our adaptive hybrid framework attains **competitive compression with stronger accuracy**, validating its effectiveness for efficient and reliable inference.

Table 6: Efficiency and compression analysis: benchmark accuracy (outside parentheses) and average thinking token length (inside parentheses) across different CoT compression methods and model variants.

| Model | MATH500 | AIME24 | AMC23 | Olympiad | LiveCodeBench | MBPP | MBPP++ | AlpacaEval2.0 | ArenaHard2.0 | Avg. |
|---|---|---|---|---|---|---|---|---|---|---|
| **1.5B Size Models** | | | | | | | | | | |
| HFT-1.5B-Think | **87.8** (4379) | 32.7 (13431) | **75.0** (10181) | 50.0 (10480) | 16.8 (13628) | **62.2** (5090) | **52.9** (5090) | **15.9** (1927) | **9.3** (10616) | **44.7** (8314) |
| HFT-1.5B | **87.8** (4330) | 32.7 (13431) | **75.0** (10181) | 48.9 (10480) | 15.7 (**7460**) | 54.8 (**2803**) | 47.4 (**2803**) | 13.1 (1101) | 6.9 (**4287**) | 42.5 (**6320**) |
| O1-Pruner | 85.4 (**2884**) | 29.3 (10655) | 69.0 (6158) | 49.7 (6568) | 16.8 (12233) | 49.7 (**711**) | 41.8 (**711**) | 13.0 (**595**) | 6.3 (3470) | 40.1 (4887) |
| DAST | 84.8 (**2428**) | 26.9 (**8429**) | 64.0 (**4637**) | 46.4 (**4968**) | 16.8 (**8238**) | 55.8 (1412) | 46.3 (1413) | 14.3 (898) | 6.0 (2760) | 40.1 (3909) |
| CoT-Valve | 87.0 (3399) | 32.0 (11078) | 62.5 (8054) | 46.1 (7827) | 17.1 (9614) | 60.1 (4042) | 51.9 (4042) | 14.1 (1491) | 8.1 (6253) | 42.1 (6200) |
| LHRMs-1.5B | **87.8** (3722) | **35.3** (**13491**) | **75.0** (9065) | **50.4** (9490) | **17.2** (9342) | 61.1 (3103) | **54.0** (3103) | 16.9 (1250) | **10.4** (5289) | **45.3** (6428) |
| **7B Size Models** | | | | | | | | | | |
| HFT-7B-Think | 93.8 (3658) | 56.7 (10778) | **95.0** (6456) | **59.7** (8376) | 38.4 (12046) | **80.3** (3251) | **68.9** (3251) | 30.6 (1731) | 23.3 (7442) | **60.7** (6332) |
| HFT-7B | 93.6 (3604) | 56.7 (10778) | **95.0** (6253) | 58.5 (**7870**) | 34.7 (7739) | 70.6 (**1658**) | 59.8 (**1658**) | 23.7 (779) | 14.0 (**2216**) | 56.4 (**4728**) |
| O1-Pruner | 93.0 (2885) | 52.0 (**9015**) | 91.0 (**4725**) | **60.8** (**5801**) | 35.1 (8351) | 79.4 (2210) | 67.9 (2210) | 24.4 (**566**) | 16.0 (2023) | 57.7 (**4198**) |
| DAST | 92.6 (**2697**) | 46.0 (**8835**) | 86.0 (**4723**) | 58.1 (4965) | 32.9 (**6045**) | 69.4 (**1074**) | 67.8 (**1074**) | 22.3 (835) | 13.8 (**1765**) | 53.3 (**3557**) |
| CoT-Valve | 93.4 (2964) | 50.0 (9692) | 93.0 (5122) | 58.5 (7151) | 37.3 (9970) | 80.1 (2987) | 67.8 (2987) | 26.5 (1553) | 23.2 (3445) | 58.9 (5097) |
| LHRMs-7B | **93.8** (**2616**) | **66.7** (11031) | **95.0** (4976) | **61.2** (7540) | **38.8** (8432) | **81.5** (1906) | **69.6** (1906) | **35.0** (1086) | **26.0** (3416) | **63.1** (4768) |

These findings highlight the practicality of our adaptive hybrid reasoning framework, which maintains strong accuracy while significantly improving inference efficiency.

### E.3 HGPO vs. Standard GRPO under Equivalent Training Resources

To verify that the improvement brought by our **Hybrid Guided Policy Optimization (HGPO)** is not due to additional computational resources, we conducted controlled reinforcement learning (RL) experiments comparing HGPO with standard GRPO under identical setups.

Both baseline models (HFT-1.5B and HFT-7B) were trained using the same RL dataset consisting of 76K queries (sourced from DeepScaleR and Tülu3) and identical hyperparameters. This ensures that any observed differences arise purely from the optimization objective rather than training cost.

As summarized in Table 7, both strategies yield similar performance on reasoning-intensive tasks. However, HGPO-trained models consistently outperform GRPO-trained ones on general-purpose benchmarks, resulting in higher overall averages. We attribute this improvement to HGPO's **adaptive selection constraint**, which promotes more effective utilization of the "no-think" mode for simple queries and reserves reasoning capacity for complex ones.

In contrast, GRPO lacks this adaptive mechanism and underutilizes the "no-think" mode during rollouts, leading to weaker general-task performance. Moreover, HGPO models achieve significantly higher **hybrid accuracy** ($H_{acc}$), reflecting stronger ability to select the appropriate reasoning mode per query.

These results confirm that adaptive thinking enhances model effectiveness **without requiring extra computational resources**, establishing HGPO as a key contributor to the efficiency and capability of hybrid reasoning systems.

Table 7: Performance comparison between HGPO and standard GRPO under equivalent training resources. $H_{acc}$ represents hybrid accuracy.

| Model | MATH500 | AIME24 | AMC23 | OlympiadBench | LiveCodeBench | MBPP | MBPP+ | AlpacaEval2.0 | ArenaHard2.0 | $H_{acc}$ | Avg. |
|---|---|---|---|---|---|---|---|---|---|---|---|
| HFT-1.5B | 87.8 | 32.7 | 75.0 | 48.9 | 15.7 | 54.8 | 47.4 | 13.1 | 6.9 | 41.4 | 42.5 |
| HFT-GRPO-1.5B | **89.0** | 34.7 | 73.1 | 50.2 | **17.9** | **61.4** | 52.9 | 14.3 | 8.1 | 47.4 | 44.6 |
| LHRMs-1.5B | 87.8 | **35.3** | **75.0** | **50.4** | 17.2 | 61.1 | **54.0** | **16.1** | **10.4** | **54.4** | **45.3** |
| HFT-7B | 93.6 | 56.7 | 95.0 | 58.5 | 34.7 | 70.6 | 59.8 | 23.7 | 14.7 | 34.2 | 56.4 |
| HFT-GRPO-7B | **95.2** | 64.0 | 94.4 | **61.5** | **39.6** | 79.6 | 67.2 | 25.8 | 18.7 | 41.9 | 60.7 |
| LHRMs-7B | 93.8 | **66.7** | **95.0** | 61.2 | 38.8 | **81.5** | **69.6** | **35.0** | **26.7** | **71.9** | **63.1** |

## E.4 Annotation Consistency Analysis for Hybrid Accuracy

To ensure the fairness and reliability of our Hybrid Accuracy ($\mathcal{H}_{acc}$) evaluation, we further computed $\mathcal{H}_{acc}$ using both GPT-4o and human annotations in addition to the Tulu-3-8B-RM reward model. For human evaluations, we randomly sampled 100 queries per model and engaged three annotators, each labeling 800 samples independently.

As shown in Table 8, the relative ranking of models based on Hybrid Accuracy remains consistent across all annotation sources (GPT-4o, human, and Tulu-3-8B-RM). All evaluation methods confirm that LHRMs-7B and LHRMs-1.5B achieve substantial improvements in selecting the correct reasoning mode.

Furthermore, we analyzed the agreement across different annotation sources, summarized in Table 9. Human-to-human agreement averaged around 60%, while human-to-RM agreement was slightly higher (62.5%). Notably, GPT-4o showed the highest consistency, comparable to agreement levels reported in MT-Bench [59] and UltraFeedback [9]. These results validate the robustness and fairness of using Tulu-3-8B-RM for hybrid accuracy evaluation.

Table 8: Hybrid Accuracy (%) of different models evaluated by Tulu-3-8B-RM, GPT-4o, and human annotators.

| Model | RM | GPT-4o | Human | Avg. |
|---|---|---|---|---|
| HFT-1.5B | 41.4 | 41.3 | 42.0 | 41.6 |
| HFT-1.5B | 48.1 | 48.2 | 46.3 | 47.5 |
| HFT-DPO-1.5B | 45.8 | 43.4 | 43.3 | 44.2 |
| LHRMs-1.5B | 54.4 | 52.8 | 53.7 | **53.6** |
| HFT-7B | 34.2 | 30.0 | 39.0 | 34.4 |
| HFT-RFT-7B | 49.7 | 48.6 | 47.7 | 48.7 |
| HFT-DPO-7B | 37.1 | 35.4 | 39.0 | 37.2 |
| LHRMs-7B | **71.9** | **71.5** | **72.0** | **71.8** |

Table 9: Agreement rates (%) between annotation sources. GPT-4o shows the highest consistency, confirming reliable alignment with human judgment.

| | GPT-4o | RM | H-1 | H-2 | H-3 | Avg. |
|---|---|---|---|---|---|---|
| GPT-4o | - | 78.5 | 69.8 | 70.1 | 69.5 | 69.8 |
| RM | - | - | 61.0 | 64.1 | 62.3 | 62.5 |
| H-1 | - | - | - | 59.6 | 61.8 | 60.7 |
| H-2 | - | - | - | - | 58.6 | 59.1 |
| H-3 | - | - | - | - | - | 60.2 |

# F  Future Work

While our current work focuses on enabling adaptive hybrid thinking based on query context, there remain several promising directions for future exploration. One potential enhancement is to allow the model to first reason about the context and complexity of the input before deciding whether to engage in extended thinking or direct response generation. This preliminary reasoning step could improve the accuracy of mode selection, leading to better balance between performance and efficiency. We plan to explore this approach in future work to further strengthen the adaptability and generality of our hybrid thinking framework.

## G  Example Outputs

**Question: What is the value of $513^2 - 487^2$? Let's think step by step and output the final answer within boxed{}. Ground Truth: 26000**

**DeepSeek-R1-Distill-Qwen-7B**: `<think>` Okay, so I need to calculate 513 squared minus 487 squared. Hmm, let me think about how to approach this. I remember that there's a formula for the difference of squares, which is a² - b² = (a - b)(a + b). Maybe I can use that here instead of calculating each square separately, which might be time-consuming. ... (3837 characters) `</think>`
To solve $513^2 - 487^2$, we use the difference of squares formula $a^2 - b^2 = (a - b)(a + b)$.
1. Identify $a = 513$ and $b = 487$. 2. Calculate $a - b$:

$$513 - 487 = 26$$

3. Calculate $a + b$:
$$513 + 487 = 1000$$

4. Multiply the results from steps 2 and 3:
$$26 \times 1000 = 26000$$

Verification by direct calculation: - $513^2 = 263169$ - $487^2 = 237169$ - Subtracting these values: $263169 - 237169 = 26000$
Both methods confirm the result.
Therefore, the final answer is $\boxed{26000}$ ✓

**LHRMs-7B**: `<no_think>` To solve $513^2 - 487^2$, we can use the difference of squares formula, which states that $a^2 - b^2 = (a+b)(a-b)$.
Here, $a = 513$ and $b = 487$.
Step 1: Calculate $a + b$:
$$513 + 487 = 1000$$

Step 2: Calculate $a - b$:
$$513 - 487 = 26$$

Step 3: Use the difference of squares formula:

$$513^2 - 487^2 = (513 + 487)(513 - 487)$$

$$513^2 - 487^2 = 1000 \times 26$$

Step 4: Perform the multiplication:
$$1000 \times 26 = 26000$$

Therefore, the final answer is: $\boxed{26000}$ `</no_think>` ✓

Figure 9: LHRMs-7B uses the No-Thinking mode to solve easy math problems.

**Question: You are given an array of integers `nums` of length `n` and a positive integer `k`. The power of an array is defined as: its maximum element if all of its elements are consecutive and sorted in ascending order, and -1 otherwise. You need to find the power of all subarrays of `nums` of size `k`. Return an integer array `results` of size `n - k + 1`, where `results[i]` is the power of `nums[i..(i + k - 1)]`.**

**HFT-7B**:

`<no_think>` To solve this problem, we need to evaluate the power of all subarrays of a given size `k` from the input array `nums`. The power of a subarray is defined as its maximum element if all elements are consecutive and sorted in ascending order; otherwise, it is -1.

Here's a step-by-step plan to solve the problem:
1. **Iterate through each subarray of size `k`**: We will slide a window of size `k` through the array `nums` and examine each subarray.
2. **Check if the subarray is sorted and consists of consecutive integers:**
  - First, check if the subarray is sorted in ascending order.
  - Then, check if the elements of the subarray are consecutive integers.

3. **Calculate the power of each subarray:** If the subarray meets the criteria, the power is the maximum element; otherwise, it is -1.

Here is the Python code that implements this logic:

```python
class Solution:
    def resultsArray(self, nums: List[int], k: int) -> List[int]:
        n = len(nums)
        result = []

        for i in range(n - k + 1):
            subarray = nums[i:i + k]
            sorted_subarray = sorted(subarray)
            max_element = subarray[-1]

            if subarray == sorted_subarray and max_element - subarray[0] == k - 1:
                result.append(max_element)
            else:
                result.append(-1)

        return result
```

`</no_think>` ✗

**LHRMs-7B**:

`<think>` Okay, I need to solve this problem where I have to find the power of all subarrays of size `k` in the given array. The power is defined as the maximum element if all elements are consecutive and sorted in ascending order. Otherwise, it's -1. Hmm, let's break down the problem. So for each subarray of length `k`, I have to check two things: whether the elements are consecutive and whether they are sorted in ascending order. ... (2645 characters) `</think>`

To solve this problem, we need to determine the power of all subarrays of a given size `k` from an array of integers. The power of a subarray is defined as the maximum element if the subarray is both sorted in ascending order and consists of consecutive integers. Otherwise, the power is -1.

**Approach**
1. **Iterate through each possible subarray**: For each starting index `i` from 0 to `'n - k'`, extract the subarray of length `k`.
2. **Check if the subarray is sorted**: Verify if each element in the subarray is less than the next element. If not, the power is -1.
3. **Check if the subarray consists of consecutive integers**: Ensure each element is exactly one more than the previous element. If not, the power is -1.
4. **Determine the power**: If both conditions are met, the power is the last element of the subarray (since it is the maximum in a sorted consecutive sequence). Otherwise, the power is -1.

```python
class Solution:
    def resultsArray(self, nums: List[int], k: int) -> List[int]:
        result = []
        n = len(nums)
        for i in range(n - k + 1):
            sub = nums[i:i+k]
            # Check if the subarray is sorted in ascending order
            is_sorted = True
            for j in range(len(sub) - 1):
                if sub[j] >= sub[j+1]:
                    is_sorted = False
                    break
            if not is_sorted:
                result.append(-1)
                continue
            # Check if the subarray consists of consecutive integers
            is_consecutive = True
            for j in range(len(sub) - 1):
                if sub[j+1] != sub[j] + 1:
                    is_consecutive = False
                    break
            if is_consecutive:
                result.append(sub[-1])
            else:
                result.append(-1)
        return result
```

✓

Figure 10: LHRMs-7B learned to select the Thinking mode to solve hard code problems.

