# OpenReview forum: "Think Only When You Need with Large Hybrid-Reasoning Models"
_NeurIPS.cc/2025/Conference — NeurIPS 2025 poster_

### Official Review · Reviewer_hw6N · 2025-06-24

**Clarity:** 3
**Significance:** 3
**Originality:** 3
**Rating:** 4
**Confidence:** 5

**Summary:**

In this paper authors introduce a new method to make reasoning models perform better on conversational benchmarks. The method consists of two stages - Hybrid Finetuning and Hybrid Group Policy Optimization. The resulting models support two generation modes - thinking and no thinking which makes them both competitive on reasoning tasks and usable in conversational scenarios.

**Questions:**

- How evaluations were done? Were all models evaluated with the same hyperparameters?
- Can authors provide evaluations on GPQA-Diamond, HumanEval, HumanEval++ for fair comparison with other LRMs?

**Ethical Concerns:**

["NO or VERY MINOR ethics concerns only"]

**Final Justification:**

I would like to thank the authors for their clarifications and for running additional experiments. My main concerns regarding additional benchmark evaluations and data decontamination were addressed therefore I will raise my score. I still think that for the fair comparison the HFT and NHFT models should have been compared to other SoTA models taking into account training compute/dataset size (e.g. DeepSeek-R1-Distill-Qwen-7B[1] was trained on 800M samples using SFT while the HFT stage only in the paper involves 1.7M samples.

[1] https://arxiv.org/pdf/2501.12948

**Limitations:**

- Only one base model was used - Qwen2.5-math-base.
- No comparison with the same compute (both training and test-time).

**Paper Formatting Concerns:**

No concerns.

**Quality:**

2

**Strengths And Weaknesses:**

### Strengths

- NHTMs outperform baselines on both reasoning and chat benchmarks.
- Good baselines - authors were comparing HGPO with both HFT and HFT+DPO/RFT.
- Only fully-accessible data and open-source models were used in this study.
- Two model scales - 1.5B and 7B.

### Weaknesses

- No comparison with DeepScaleR-1.5B-Preview[1] although the DeepScaleR data was used.
- No evaluations on GPQA-Diamond[2], HumanEval, HumanEval++.
- No information on training data decontamination. E.g. AIME dataset[3] might contain samples from AIME24 benchmark.
- Grammar errors ("to lean a effective NHTMs").

### References

1. https://huggingface.co/agentica-org/DeepScaleR-1.5B-Preview
2. https://openreview.net/forum?id=Ti67584b98
3. https://huggingface.co/datasets/di-zhang-fdu/AIME_1983_2024

---

> ### Author Rebuttal · Authors · 2025-07-31
>
> We are grateful to the Reviewer hw6N for the extensive review. We address your questions point by point below.
>
> ---
> > **Q1:** How evaluations were done? Were all models evaluated with the same hyperparameters?
>
> **A1**: Thank you for raising this question. We performed rigorous evaluations on our models across various benchmarks, **ensuring that both NHTMs and baseline models were evaluated under the same experimental setup to guarantee a fair comparison**. As detailed in Appendix D ("Evaluation Settings"), we consistently used a temperature of 0.6, a top-p value of 0.95, and a maximum generation length of 35,000 tokens for all models across all tasks. To ensure the stability and reliability of our results, **the reported scores for all reasoning tasks are the averages of five independent runs**.
>
> For specific benchmark suites, we adhered to widely-accepted evaluation frameworks. MATH500, AIME24, and LiveCodeBench were evaluated using the OpenR1 evaluation framework, while MBPP and MBPP+ were evaluated using the EvalPlus framework. Further details are provided in Appendix D.
>
> ---
> > **Q2:** Can authors provide evaluations on GPQA-Diamond, HumanEval, HumanEval++ for fair comparison with other LRMs?
>
> **A2:** Thank you for the valuable suggestion. We have conducted these additional evaluations to ensure a fair and comprehensive comparison at **Table A** and **Table B**. **Our experimental results demonstrate that our NHTMs-1.5B and NHTMs-7B models outperform the DeepSeek-R1-Distill-Qwen counterparts on both HumanEval and HumanEval++ benchmarks, and achieves the highest average performance among all compared models**.
>
> Although our NHTMs-7B performs slightly lower on GPQA-Diamond compared to DeepSeek-R1-Distill-Qwen-7B, it still shows significant improvements over the HFT-1.5B baseline. This is particularly noteworthy because our training dataset explicitly excluded Science-related data, which is primarily the domain covered by the GPQA-Diamond benchmark (as illustrated in Appendix C, Table 2 and Table 3). Therefore, the observed performance indicates that our Hybrid Group Policy Optimization (HGPO) method effectively enhances generalization, allowing the model to achieve robust performance even on tasks outside its direct training distribution.
>
> In summary, the additional evaluations further validate the effectiveness and generalization capability of our proposed model.
>
> ***Table A**: Performance Comparison on Comprehensive Benchmarks (1.5B size)*
>
> | Model                          | HumanEval | HumanEval++ | GPQA-Diamond | MATH-500 | AIME 2024 | AMC23 | Olympiad | LiveCode-Bench | MBPP | MBPP+ | Alpaca-Eval2.0 | Arena-Hard v2 | Avg |
> |-------------------------------|-----------|-------------|---------------|----------|------------|-------|----------------|----------------|------|--------|----------------|---------------|---------|
> | DeepSeek-R1-Distill-Qwen-1.5B | 67.7      | 59.8        | **33.8**      | 83.9     | 28.9       | 62.9  | 43.3           | 16.8           | 54.2 | 46.3   | 5.6            | 2.7           | 42.2    |
> | HFT-1.5B                      | 65.9      | 62.2        | 27.3          | **87.8** | 32.7       | **75.0** | 48.9           | 15.7           | 54.8 | 47.4   | 13.1           | 6.9           | 44.8    |
> | NHTMs-1.5B                    | **75.0**  | **70.7**    | 31.3          | **87.8** | **35.3**   | **75.0** | **50.4**       | **17.2**       | **61.1** | **54.0** | **16.1**       | **10.4**      | **48.7** |
>
> ***Table B**: Performance Comparison on Comprehensive Benchmarks (7B size)*
> | Model                          | HumanEval | HumanEval++ | GPQA-Diamond | MATH-500 | AIME 2024 | AMC23 | Olympiad | LiveCode-Bench | MBPP | MBPP+ | Alpaca-Eval2.0 | Arena-Hard v2 | Avg|
> |-------------------------------|-----------|-------------|---------------|----------|------------|-------|----------------|----------------|------|--------|----------------|---------------|---------|
> | DeepSeek-R1-Distill-Qwen-7B   | 88.4      | 81.7        | **49.1**      | 92.8     | 55.5       | 91.5  | 58.1           | 37.6           | 74.3 | 64.3   | 19.1           | 17.9          | 60.9    |
> | HFT-7B                        | 84.8      | 79.3        | 31.3          | 93.6     | 56.7       | **95.0** | 58.5           | 34.7           | 70.6 | 59.8   | 23.7           | 14.7          | 58.6    |
> | NHTMs-7B                      | **89.6**  | **82.9**    | 41.4          | **93.8** | **66.7**   | **95.0** | **61.2**       | **38.8**       | **81.5** | **69.6** | **35.0**       | **26.7**      | **65.2** |
>
> ---
> > **W1:** No comparison with DeepScaleR-1.5B-Preview[1] although the DeepScaleR data was used.
>
> **A3**: Thank you for your valuable feedback. We have now included the results of DeepScaleR-1.5B-Preview in the comparison table below, alongside NHTMs-1.5B for a fair evaluation across key benchmarks:
>
> | Model                         | MATH500 | AIME24 | AMC23 | Olympiad | MBPP | MBPP+ | Alpaca-Eval2.0 | ArenaHard v2 | Avg  |
> |-------------------------------|---------|--------|-------|----------|--------|-------|----------------|--------------|------|
> | DeepSeek-R1-Distill-Qwen-1.5B | 83.9    | 28.9   | 62.9  | 43.3     | 54.2 | 46.3  | 5.6            | 2.7          | 41.0 |
> | DeepScaleR-1.5B-Preview       | 87.8    | **43.1**   | 73.6  | 50.0     | 56.6 | 46.6  | 6.4            | 6.3          | 46.2 |
> | NHTMs-1.5B                     | **87.8**    | 35.3   | **75.0**  | **50.4**     | **61.1** | **54.0**  | **16.9**           | **10.4**         | **48.8** |
>
> As shown in the table, **our NHTMs-1.5B model achieves the highest average performance among all compared models, outperforming both DeepScaleR-1.5B-Preview and DeepSeek-R1-Distill-Qwen-1.5B**.
>
> In particular, NHTMs-1.5B achieves comparable results to DeepScaleR-1.5B-Preview on math-focused tasks such as MATH500, and demonstrates clear advantages on tasks requiring adaptive reasoning, including MBPP and AlpacaEval2.0. Although it underperforms on AIME24, we attribute this to the difference in training budgets—DeepScaleR was trained for 1.7k steps, whereas our model was only trained for fewer than 500 steps in this comparison.
>
> These results highlight the efficiency and generalization capability of our hybrid-thinking framework, even under limited training. We appreciate your suggestion and will put DeepScaleR-1.5B-Preview as a comparison point in next version of our paper to provide a more comprehensive evaluation.
>
> ---
> > **W2:** No information on training data decontamination. E.g. AIME dataset[3] might contain samples from AIME24 benchmark.
>
> **A4:** Thank you for pointing this out. Although we briefly mentioned the removal of overlapping samples in Section 2.2 ("Data Construction", line 96), we agree that a more detailed explanation of our decontamination process is necessary.
>
> To ensure the fairness of our evaluation, **we uniformly applied the Simhash algorithm [1] to calculate the similarity between each training sample and all samples across our evaluation benchmarks. Any instances with a similarity score greater than 0.8 were removed to strictly prevent data contamination**.
>
> Specifically regarding the AIME dataset, we took extra precautionary measures to avoid any temporal or content overlap with the AIME24 benchmark:
>
> * **Source Filtering**: Our training data only included AIME problems from 2023 and earlier, thereby ensuring there was no temporal overlap with the AIME24 benchmark at the source level.
>
> * **Precise Deduplication**: In addition to source filtering, we applied the aforementioned Simhash deduplication process against the AIME24 test set as an additional safeguard. As reflected in Table 2 of our paper, the initial AIME dataset contained 933 problems. After applying this rigorous decontamination process, 741 problems were retained for training, which further confirms the execution of our decontamination procedure.
>
> Through these two measures—source control and precise deduplication—we have ensured there is no data leakage from the AIME24 benchmark. We will add a more detailed description of our decontamination process to the appendix in the final version to improve the transparency and reproducibility of our work.
>
> **Reference**
>
> [1] Detecting near-duplicates for web crawling. Manku, Gurmeet Singh and Jain, Arvind and Das Sarma, Anish.
>
> ---
> > **W3:** Grammar errors ("to lean a effective NHTMs").
>
> **A5:** Thank you for pointing out the grammatical error, and we will correct the grammar in the latest version of the paper.
>
> ----
> We want to express our sincere gratitude for your review. If you have any further questions or concerns, please feel free to contact us at any time. We are always available and look forward to further discussions with you. :)
>
> Best regards,
>
> All Authors

---

> ### Author Response · Authors · 2025-08-04
> **Official Comment by Authors**
>
> Dear Reviewer hw6N,
>
> Thank you for taking the time to engage with our rebuttal and provide valuable feedback. We sincerely appreciate your efforts in considering our responses and the additional experiments.
>
> We hope that the supplementary experiments and clarifications we have provided address all your concerns. If any issues remain unresolved or if you have further questions, please feel free to contact us at any time. We would be happy to engage in further discussion and provide any additional clarifications as needed. :)
>
> Thank you once again for your time and effort in reviewing our work.
>
> Best regards,
>
> All Authors

---

> > ### Comment · Reviewer_hw6N · 2025-08-05
> >
> > My main concerns regarding additional benchmark evaluations and data decontamination were addressed therefore I will raise my score. Thank you.

---

> > > ### Author Response · Authors · 2025-08-06
> > > **Official Comment by Authors**
> > >
> > > Thank you very much for your positive response. We are very glad that your main concerns have been resolved, and we truly appreciate your decision to raise the score.
> > >
> > > We are grateful for your thoughtful feedback and for engaging with our responses throughout the rebuttal process. Thank you again for your time and consideration.
> > >
> > > Best regards,
> > >
> > > All Authors

---

### Official Review · Reviewer_ofyf · 2025-06-27

**Clarity:** 3
**Significance:** 3
**Originality:** 3
**Rating:** 5
**Confidence:** 4

**Summary:**

This paper proposes a complete pipeline to build a native hybrid thinking LLM which can decide whether conduct long CoT thinking according to the difficulty of the problem by themselves. The entire pipeline follows a standard SFT-then-RL pattern. The SFT stage integrates both reasoning-intensive (Thinking) and direct-answer (No-Thinking) data to establish a robust initialization for following RL training. In RL training phase, this paper proposes a novel HGPO and delicate reward functions to enable the LLM to internally choose the thinking mode when dealing problems with different difficulties. Extensive experiments demonstrate that NHTMs not only possess strong reasoning capabilities but also can largely save computation resources while dealing simple problems.

**Questions:**

1. In Equation (10), I am particularly interested in the term “GRPO for inter-group advantage $A_{\text{inter}}$.” However, I do not fully understand how this term effectively contributes to the decision of whether to think or not. I would appreciate it if the authors could provide a more detailed explanation. It may also be helpful for other readers if the authors include additional clarification or discussion of this term in the paper.

**Ethical Concerns:**

["NO or VERY MINOR ethics concerns only"]

**Final Justification:**

All of my concerns have been well addressed.

**Limitations:**

Yes.

**Paper Formatting Concerns:**

No major formatting issues.

**Quality:**

3

**Strengths And Weaknesses:**

Strengths:
1. The overall writing of this paper is excellent, and the logic is easy to follow.
2. The proposed training pipeline is well-designed, and the final experimental results are solid, comprehensive, and convincing.

Weaknesses:
This paper does not have any significant weaknesses. Here are some of my suggestions:
1. In Figure 5, the y-axis label for "Thinking Ratio" appears to be missing.
2. A key contribution of the proposed NHTMs is their ability to autonomously decide whether to think, thereby saving computational budget. It would be more intuitive to include the average number of output tokens for both the baselines and NHTMs across all benchmarks.
3. In my understanding, Hybrid Accuracy is a model-specific metric, as $m_\text{gt}$ is determined by the evaluated LLM. If multiple NHTMs are evaluated, the proposed Hybrid Accuracy metric may not fairly compare them.

---

> ### Author Rebuttal · Authors · 2025-07-31
>
> > **Q1.**: In Equation (10), I am particularly interested in the term “GRPO for inter-group advantage .” However, I do not fully understand how this term effectively contributes to the decision of whether to think or not. I would appreciate it if the authors could provide a more detailed explanation. It may also be helpful for other readers if the authors include additional clarification or discussion of this term in the paper.
>
> **A1.** Thank you for your valuable suggestion. We agree that the “GRPO for inter-group advantage” term in Equation (10) plays a crucial role and deserves a more detailed explanation. To clarify, the process unfolds as follows:
>
> 1. **Scoring responses (Eq. 7):** For each query, we sample both *Thinking* and *No-Thinking* responses, and score them using the reward model.
> 2. **Computing average rewards (Eq. 8):** We then compute the average reward for both reasoning modes across these samples.
> 3. **Selecting the better mode:** By comparing the two average rewards, we identify which reasoning mode—*Thinking* or *No-Thinking*—is more appropriate for the given query.
> 4. **Assigning learning signals (Eq. 9):** We then get inter-group reward by assigning a high reward (1) to responses sampled from the better mode and a low reward (0) to those from the less effective mode.
> 5. **Learning via GRPO (Eq. 10):** The “GRPO for inter-group advantage” then learns from this inter-group reward contrast, encouraging the model to favor the reasoning mode that is more beneficial for a specific query.
>
> We will revise the manuscript to enhance the explanation of this term and its role in reasoning-mode selection, as we believe this will indeed help readers better understand our method. Thank you again for your insightful comment.
>
> > **W1:** In Figure 5, the y-axis label for "Thinking Ratio" appears to be missing.
>
> **A2**：Thank you for your careful review and for pointing out the issue with Figure 5. We agree with your observation — the "Thinking Ratio" and "Accuracy" currently share the same y-axis, but the label for "Thinking Ratio" was inadvertently omitted. This was an oversight on our part. In the revised version of the paper, we will explicitly add the missing label to ensure clarity and avoid any potential confusion for readers. We appreciate your attention to detail in helping us improve the quality of our paper.
>
> > **W2:** A key contribution of the proposed NHTMs is their ability to autonomously decide whether to think, thereby saving computational budget. It would be more intuitive to include the average number of output tokens for both the baselines and NHTMs across all benchmarks.
>
> **A3:** As suggested, we have provided a detailed comparison of the output lengths across all benchmarks in **Tables 1 and 2**, alongside the corresponding accuracy results. In each cell, the number in parentheses indicates the average number of output tokens. These tables cover the proposed NHTMs (Hybrid-Thinking Models), the full-thinking baselines (HFT-1.5B-Think and HFT-7B-Think), and the DeepSeek-R1-Distill-Qwen models.
>
> * Across most benchmarks, the hybrid-thinking models consistently produce shorter outputs while maintaining equal or better performance, demonstrating their ability to reduce unnecessary reasoning for simpler queries and thus improve inference efficiency.
> * On the more challenging AIME24 benchmark, the hybrid-thinking models generate longer outputs compared to both the thinking-only and DeepSeek models, as they engage in more elaborate reasoning to solve complex problems. This additional reasoning leads to significantly improved performance—NHTMs achieve the best accuracy at both the 1.5B and 7B model sizes, with the 7B variant outperforming DeepSeek-R1-Distill-Qwen by **8** percentage points.
>
> These results clearly demonstrate that our hybrid-thinking model achieves improved inference efficiency by adaptively reducing unnecessary reasoning steps for simpler queries.
>
> ***Table 1*. Accuracy and Output Length Comparison on Various Benchmarks (1.5B size)**
>
> | **Model**                      | MATH500           | AIME24            | AMC23             | Olympiad          | LiveCodeBench     | MBPP             | MBPP++           | AlpacaEval2.0     | ArenaHard2.0       | **Avg.**        |
> |-------------------------------|-------------------|-------------------|-------------------|-------------------|--------------------|------------------|------------------|--------------------|---------------------|------------------|
> | DeepSeek-R1-Distill-Qwen-1.5B | 83.9 (4059)       | 28.9 (13366)      | 62.9 (9556)       | 43.3 (10921)      | 16.8 (13002)       | 54.2 (4146)      | 46.3 (4146)      | 5.6 (4090)         | 2.7 (8285)          | 38.3 (7952)       |
> | HFT-1.5B-Think                | 87.8 (4379)   | 32.7 (13431)      | 75.0 (10181)      | 50.0 (10480)      | 16.8 (13628)       | **62.2** (5090)  | 52.9 (5090)      | 15.9 (1927)        | 9.3 (10616)         | 44.7 (8314)       |
> | HFT-1.5B                     | 87.8 (4330)   | 32.7 (13431)      | 75.0 (10181)      | 48.9 (10480)      | 15.7 (**7460**) | 54.8 (**2803**)  | 47.4 (**2803**)  | 13.1 (**1101**)    | 6.9 (**4287**)      | 42.5 (**6320**)   |
> | NHTMs-1.5B                   | **87.8** (**3722**) | **35.3** (13491)  | **75.0** (**9065**) | **50.4** (**9490**) | **17.2** (9342)    | 61.1 (3103)      | **54.0** (3103)  | **16.9** (1250)     | **10.4** (5289)     | **45.3** (6428)   |
>
> ***Table 2*. Accuracy and Output Length Comparison on Various Benchmarks (7B size)**
>
> | **Model**                      | MATH500           | AIME24             | AMC23              | Olympiad           | LiveCodeBench     | MBPP              | MBPP++            | AlpacaEval2.0      | ArenaHard2.0        | **Avg.**        |
> |-------------------------------|-------------------|--------------------|--------------------|--------------------|--------------------|-------------------|-------------------|---------------------|----------------------|------------------|
> | DeepSeek-R1-Distill-Qwen-7B   | 92.8 (3558)       | 55.5 (9488)        | 91.5 (6255)        | 58.1 (8635)        | 37.6 (11669)       | 74.3 (2824)       | 64.3 (2824)       | 19.1 (2209)         | 17.9 (5282)          | 56.8 (5860)       |
> | HFT-7B-Think                  | 93.8 (3658)   | 56.7 (10778)       | 95.0 (6456)    | 59.7 (8376)        | 38.4 (12046)       | 80.3 (3251)       | 68.9 (3251)       | 30.6 (1731)         | 23.3 (7442)          | 60.7 (6332)       |
> | HFT-7B                        | 93.6 (3604)       | 56.7 (10778)       | 95.0 (6253)    | 58.5 (7870)        | 34.7 (**7739**) | 70.6 (**1658**)   | 59.8 (**1658**)   | 23.7 (**779**)       | 14.0 (**2216**)      | 56.4 (**4728**)   |
> | NHTMs-7B                      | **93.8** (**2616**) | **66.7** (11031) | **95.0** (**4976**) | **61.2** (**7540**) | **38.8** (8432) | **81.5** (1906) | **69.6** (1906) | **35.0** (1086) | **26.0** (3416) | **63.1** (4768) |
>
>
> > **W3:** In my understanding, Hybrid Accuracy is a model-specific metric, as $m_{gt}$
>  is determined by the evaluated LLM. If multiple NHTMs are evaluated, the proposed Hybrid Accuracy metric may not fairly compare them.
>
> **A4:** Thank you for the insightful comment. We agree with your observation that Hybrid Accuracy is indeed a model-specific metric, as it directly depends on the selection strategy employed by the evaluated hybrid model.
>
> We would like to clarify that Hybrid Accuracy is not intended as an absolute metric to evaluate overall model performance across different models. Instead, it specifically reflects the effectiveness of a given model in making correct choices between the "thinking" and "no-thinking" modes within its own decision-making mechanism.
>
> For example, consider a hypothetical hybrid model that always performs better in the "thinking" mode compared to the "no-thinking" mode (perhaps due to insufficient training in the "no-thinking" mode). If this model always selects "thinking," it will achieve a high Hybrid Accuracy score, but this does not necessarily indicate that the model itself has superior overall capabilities—it merely signifies that it correctly identifies its optimal internal mode.
>
> Conversely, if this model mistakenly prefers the "no-thinking" mode under similar conditions, the Hybrid Accuracy score will be low, clearly indicating poor internal decision-making. Thus, Hybrid Accuracy primarily assesses the model’s internal decision logic rather than its absolute performance on tasks.
>
> In the revised version of our paper, we will explicitly clarify this aspect and provide illustrative examples to better convey the scope and intention of the Hybrid Accuracy metric.
>
> ---
>
> We sincerely thank the Reviewer ofyf for the thoughtful comments and constructive suggestions :). Your insights have been invaluable in improving the clarity and completeness of our work.
>
> If you have any further questions or concerns, please feel free to contact us at any time. We are always available and look forward to further discussions with you. :)
>
> Best regards,
>
> All Authors

---

> > ### Comment · Reviewer_ofyf · 2025-08-04
> >
> > Thank you for your responses! All of my concerns have been well addressed.

---

> ### Author Response · Authors · 2025-08-04
> **Official Comment by Authors**
>
> Thank you very much for your positive feedback! We are thrilled to hear that all of your concerns have been well addressed. We will take your advice into account in the latest version to further improve the clarity and quality of our work.
>
> We truly appreciate your time and thoughtful review.
>
> Best regards,
>
> All Authors

---

### Official Review · Reviewer_jtLz · 2025-07-02

**Clarity:** 4
**Significance:** 3
**Originality:** 4
**Rating:** 5
**Confidence:** 5

**Summary:**

This paper presents Native Hybrid Thinking Models, which adaptively decide whether to engage in extended thinking based on user query context, addressing the overthinking issue in Large Reasoning Models. It proposes a two-stage training pipeline: Hybrid Fine-Tuning (HFT) for cold start and Hybrid Group Policy Optimization for learning to select appropriate thinking modes, along with a Hybrid Accuracy metric to assess hybrid thinking capability. Experiments show Native Hybrid Thinking Models outperform LLMs in reasoning and general capabilities while improving efficiency by reducing unnecessary thinking for simple queries.

**Questions:**

1. It is surprising that adaptive thinking shows significantly greater improvement on AIME 2024 than on Math500. Intuitively, simple problems' thinking process should be easily compressible, while difficult problems' thinking process should not be compressed or should even benefit from length scaling to improve performance. Could you compare the thinking ratio and thinking token length changes between MATH500/AIME 2024 to explain this phenomenon?

**Ethical Concerns:**

["NO or VERY MINOR ethics concerns only"]

**Final Justification:**

The author has addressed my concerns regarding the baseline setup, the detailed think ratio statistics for different difficulty levels, and the token length statistics. Therefore, my confidence in supporting the acceptance of this paper has increased.

**Limitations:**

yes

**Quality:**

3

**Strengths And Weaknesses:**

**Strengths**
1. Making reasoning models more efficient is an important research problem.
2. The paper proposes a reasonable approach (including the complete pipeline of SFT and RL) with certain novelty.
3. The paper is well-written and easy to follow.

**Weakness**
1. Direct comparison with DeepSeek-R1-Distill-Qwen-7B/1.5B & NHTMs-7B/1.5B is insufficient to demonstrate the method's effectiveness. Based on Stage I, training a baseline model using the same RL data (76K queries from Deepscaler and Tülu3) with standard GRPO to achieve high performance would better illustrate whether implementing adaptive thinking compromises the model's capability ceiling. Simply comparing against DeepSeek-R1-Distill-Qwen-7B/1.5B is inadequate, as you have expended additional computational resources that might enable the model to reach a significantly higher ceiling, making adaptive CoT no longer cost-effective. This point requires experimental validation.
2. As research on adaptive CoT, reporting the thinking token length statistics for each dataset and comparing with CoT compression works [1][2][3] would strengthen the contribution of this work.
3. Llama-3.1-Tulu-3-8B-RM is used both for RL training and for computing Hybrid Accuracy in Section 2.4. Could this lead to unfair Hybrid Accuracy evaluation?

[1] Luo H, Shen L, He H, et al. O1-Pruner: Length-Harmonizing Fine-Tuning for O1-Like Reasoning Pruning[J]. arXiv preprint arXiv:2501.12570, 2025.

[2] Ma X, Wan G, Yu R, et al. CoT-Valve: Length-Compressible Chain-of-Thought Tuning[J]. arXiv preprint arXiv:2502.09601, 2025.

[3] Shen Y, Zhang J, Huang J, et al. Dast: Difficulty-adaptive slow-thinking for large reasoning models[J]. arXiv preprint arXiv:2503.04472, 2025.

---

> ### Author Rebuttal · Authors · 2025-07-31
>
> We are grateful to the Reviewer for the extensive review. We address your questions point by point below.
>
> > **Q1:** It is surprising that adaptive thinking shows significantly greater improvement on AIME 2024 than on Math500. Could you compare the thinking ratio and thinking token length changes between MATH500/AIME 2024 to explain this phenomenon?
>
> **A1:** We compared the thinking ratio and average reasoning length across MATH500 and AIME24 (see Table 1). The results are consistent with the intuition that simple problems benefit more from reasoning compression, while difficult problems require full reasoning.
>
> On MATH500, which contains many relatively simple problems, NHTMs significantly reduce both the thinking ratio and the average reasoning length, indicating that the model effectively skips unnecessary reasoning by selecting the No-Thinking mode. This improves efficiency without sacrificing accuracy. In contrast, on the more challenging AIME24, the model applies 100% Thinking with slightly longer reasoning chains, reflecting its consistent use of detailed step-by-step reasoning where it is truly needed.
>
> Together, these trends explain the greater performance gains on AIME24: the model allocates full reasoning to difficult tasks while leveraging No-Thinking to bypass reasoning for simpler ones.
>
>
> **Table 1: Comparison of thinking ratio (%) and average reasoning length (tokens) between HFT and NHTMs on MATH500 and AIME24.**
>
> |Model|MATH500 (Think Ratio)|MATH500 (Length)|AIME24 (Think Ratio)|AIME24 (Length)|
> |-|-|-|-|-|
> |HFT-1.5B|99.2|4330|100|13431|
> |NHTMs-1.5B|90.2|3722|100|13491|
> |HFT-7B|98.2|3604|100|10778|
> |NHTMs-7B|75.2|2616|100|11031|
>
> ---
> > **W1**: Direct comparison with DeepSeek-R1-Distill-Qwen-7B/1.5B & NHTMs-7B/1.5B is insufficient to demonstrate the method's effectiveness....as you have expended additional computational resources that might enable the model to reach a significantly higher ceiling, making adaptive CoT no longer cost-effective. This point requires experimental validation.
>
> **A2:** We thank the reviewer for pointing out the need to isolate the effect of adaptive thinking from computational budget. To address this, we conducted additional experiments where baseline models (HFT-1.5B and HFT-7B) were trained with standard GRPO using the exact same RL dataset (76K queries from DeepScaleR and Tülu3) and training setup as our HGPO method.
>
> As summarized in Table B, both training strategies yield similar performance on reasoning-intensive tasks. However, HGPO-trained models consistently outperform GRPO-trained counterparts in general tasks, leading to higher overall performance. We attribute this to HGPO’s adaptive selection constraint, which encourages more effective use of the "no-think" mode on simpler queries, and preserves reasoning capacity for complex problems. In contrast, GRPO lacks this adaptive mechanism and underutilizes the "no-think" mode during rollouts, limiting gains on general tasks.
>
> Moreover, HGPO models achieve significantly higher hybrid accuracy ($H_{\text{acc}}$), demonstrating stronger ability to select the appropriate reasoning mode per query. This confirms that adaptive thinking not only avoids additional computational cost, but also enhances the model's effectiveness across diverse tasks.
>
> These results validate the value of HGPO in building a more efficient and capable hybrid reasoning system under equivalent training resources.
>
> **Table B: Performance Comparison of Standard GRPO vs. HGPO**
>
> |Model|MATH-500|AIME 2024|AMC23|Olympiadbench|LiveCodeBench|MBPP|MBPP+|AlpacaEval2.0|ArenaHard2.0|$H_{acc}$|Avg.|
> |-|-|-|-|-|-|-|-|-|-|-|-|
> |HFT-1.5B|87.8|32.7|75.0|48.9|15.7|54.8|47.4|13.1|6.9|41.4|42.5|
> |HFT-GRPO-1.5B|**89.0**|34.7|73.1|50.2|**17.9**|**61.4**|52.9|14.3|8.1|47.4|44.6|
> |NHTMs-1.5B|87.8|**35.3**|**75.0**|**50.4**|17.2|61.1|**54.0**|**16.1**|**10.4**|**54.4**|**45.3**|
> |HFT-7B|93.6|56.7|95.0|58.5|34.7|70.6|59.8|23.7|14.7|34.2|56.4|
> |HFT-GRPO-7B|**95.2**|64.0|94.4|**61.5**|**39.6**|79.6|67.2|25.8|18.7|41.9|60.7|
> |NHTMs-7B|93.8|**66.7**|**95.0**|61.2|38.8|**81.5**|**69.6**|**35.0**|**26.7**|**71.9**|**63.1**|
>
> ---
> > **W2**: Reporting the thinking token length statistics for each dataset and comparing with CoT compression works would strengthen the work.
>
> **A2:** We thank the reviewer for the insightful suggestion on analyzing thinking token lengths and comparing with existing CoT compression methods [1–3]. In response, we present detailed results in Table C, including average reasoning lengths and corresponding accuracies:
>
> * Compared to the full-length thinking baselines (HFT-1.5B-Think and HFT-7B-Think), our NHTMs significantly reduce token lengths across most tasks while achieving comparable or better accuracy. This confirms the efficiency and effectiveness of our adaptive reasoning strategy.
>
> * When compared with prior CoT compression methods—O1-Pruner, CoT-Valve, and DAST—our approach achieves a better balance between compression and accuracy. Although some methods (e.g., O1-Pruner, DAST) yield shorter outputs, they often suffer from performance drops. In contrast, NHTMs maintain or improve accuracy with meaningful reductions in reasoning length.
>
> These results validate the practical advantage of our adaptive CoT framework for efficient and reliable inference. We wil add these results into our newest version.
>
> **Table C. Comparison of Accuracy and Average Thinking Token Length (in Parentheses) Across CoT Compression Methods and Model Variants**
>
> |**Model**|MATH500|AIME24|AMC23|Olympiad|LiveCodeBench|MBPP|MBPP++|AlpacaEval2.0|ArenaHard2.0|**Avg**|
> |-|-|-|-|-|-|-|-|-|-|-|
> |**1.5B Size Models**|||||||||||
> |HFT-1.5B-Think|87.8 (4379)|32.7 (13431)|75.0 (10181)|50.0 (10480)|16.8 (13628)|**62.2** (5090)|52.9 (5090)|**15.9** (1927)|9.3 (10616)|44.7 (8314)|
> |HFT-1.5B|87.8 (4330)|32.7 (13431)|75.0 (10181)|48.9 (10480)|15.7 (**7460**)|54.8 (2803)|47.4 (2803)|13.1 (1101)|6.9 (4287)|42.5 (6320)|
> |O1-Pruner|85.4 (2884)|29.3 (10655)|69.0 (6158)|49.7 (6568)|16.8 (12233)|49.7 (**711**)|41.8 (**711**)|13.0 (**595**)|6.3 (3470)|40.1 (4887)|
> |DAST|84.8 (**2428**)|26.9 (**8429**)|64.0 (**4637**)|46.4 (**4968**)|16.8 (8238)|55.8 (1412)|46.3 (1413)|14.3 (898)|6.0 (**2760**)|40.1 (**3909**)|
> |CoT-Valve|87.0 (3399)|32.0 (11078)|62.5 (8054)|46.1 (7827)|17.1 (9614)|60.1 (4042)|51.9 (4042)|14.1 (1491)|8.1 (6253)|42.1 (6200)|
> |NHTMs-1.5B|**87.8** (3722)|**35.3** (13491)|**75.0** (9065)|**50.4** (9490)|**17.2** (9342)|61.1 (3103)|**54.0** (3103)|16.9 (1250)|**10.4** (5289)|**45.3** (6428)|
> |**7B Size Models**|||||||||||
> |HFT-7B-Think|93.8 (3658)|56.7 (10778)|95.0 (6456)|59.7 (8376)|38.4 (12046)|80.3 (3251)|68.9 (3251)|30.6 (1731)|23.3 (7442)|60.7 (6332)|
> |HFT-7B|93.6 (3604)|56.7 (10778)|95.0 (6253)|58.5 (7870)|34.7 (7739)|70.6 (1658)|59.8 (1658)|23.7 (779)|14.0 (2216)|56.4 (4728)|
> |O1-Pruner|93.0 (2885)|52.0 (9015)|91.0 (4725)|60.8 (**5801**)|35.1 (8351)|79.4 (2210)|67.9 (2210)|24.4 (**566**)|16.0 (2023)|57.7 (4198)|
> |DAST|92.6 (2697)|46.0 (**8835**)|86.0 (**4723**)|58.1 (4965)|32.9 (**6045**)|69.4 (**1074**)|58.9 (**1074**)|22.3 (835)|13.8 (**1765**)|53.3 (**3557**)|
> |CoT-Valve|93.4 (2964)|50.0 (9692)|93.0 (5122)|58.5 (7151)|37.3 (9970)|80.1 (2987)|67.8 (2987)|26.5 (1553)|23.2 (3445)|58.9 (5097)|
> |NHTMs-7B|**93.8** (**2616**)|**66.7** (11031)|**95.0** (4976)|**61.2** (7540)|**38.8** (8432)|**81.5** (1906)|**69.6** (1906)|**35.0** (1086)|**26.0** (3416)|**63.1** (4768)|
>
> ---
> > **W3**: Tulu-3-8B-RM is used both for RL training and for computing Hybrid Accuracy. Could this lead to unfair Hybrid Accuracy evaluation?
>
> We appreciate the reviewer’s thoughtful question. To ensure the fairness and reliability of our Hybrid Accuracy (HAcc) evaluation, we have additionally computed HAcc using both GPT-4o annotations and human annotations. For the human annotations, we sampled 100 queries per model and engaged three annotators, each labeling 800 samples independently.
>
> As presented in Table D, the results indicate that the relative ordering (partial ordering) of models based on Hybrid Accuracy remains consistent across annotations derived from GPT-4o, human evaluators, and Tulu-3-8B-RM. All annotation methods confirm that NHTMs-7B and NHTMs-1.5B achieve substantial improvements in selecting the correct reasoning mode.
>
> Furthermore, we conducted an agreement analysis across these annotation approaches (see Table E). The average agreement between human annotators was 60%, while the agreement between human annotators and Tulu-3-8B-RM was slightly higher at 62.5%. Notably, GPT-4o demonstrated the highest agreement rates, comparable to previously reported agreement levels in MT-Bench [4] and UltraFeedback [5]. These results collectively validate the reliability and fairness of employing Tulu-3-8B-RM for Hybrid Accuracy evaluation.
>
> **Table D: Hybrid Accuracy (%) of Different Models**
>
> |Model|Tulu-3-8B-RM|GPT-4o|Human|Avg|
> |-|-|-|-|-|
> |1.5B-SFT|41.4|41.3|42.0|41.6|
> |HFT-1.5B|48.1|48.2|46.3|47.5|
> |HFT-DPO-1.5B|45.8|43.4|43.3|44.2|
> |NHTMs-1.5B|54.4|52.8|53.7|53.6|
> |HFT-7B|34.2|30.0|39.0|34.4|
> |HFT-RFT-7B|49.7|48.6|47.7|48.7|
> |HFT-DPO-7B|37.1|35.4|39.0|37.2|
> |NHTMs-7B|71.9|71.5|72.0|71.8|
>
>
> **Table E: Agreement Rates (%) Between Annotators**
>
> ||GPT-4o|RM|H-1|H-2|H-3|Average|
> |-|-|-|-|-|-|-|
> |GPT-4o|-|78.5%|69.8%|70.1%|69.5%|69.8%|
> |RM|-|-|61.0%|64.1%|62.3%|62.5%|
> |H-1|-|-|-|59.6%|61.8%|60.7%|
> |H-2|-|-|-|-|58.6%|59.1%|
> |H-3|-|-|-|-|-|60.2%|
>
> [4] Cui, Ganqu, et al. "Ultrafeedback: Boosting language models with scaled ai feedback."
>
> [5] Xu, Jiazheng, et al. "Imagereward: Learning and evaluating human preferences for text-to-image generation."
>
> ---
> We sincerely thank the Reviewer jtLz for the thoughtful comments and constructive suggestions :). If you have any further questions or concerns, please feel free to contact us at any time.
>
> Best regards,
>
> All Authors

---

> ### Comment · Reviewer_jtLz · 2025-08-06
> **Response to Authors**
>
> Thank you for the detailed experimental supplements provided by the authors. Most of my concerns have been resolved. Based on this, I will increase my confidence score and support the acceptance of this paper.
>
> However, I have an additional question. According to your response to Q1, the Think Ratio for AIME2024 is 100%, which means the model does not switch reasoning mode per query when solving AIME2024 problems. Given this, why does NHTMs-7B still show improvements over HFT-GRPO-7B on AIME2024 in Table B?

---

### Official Review · Reviewer_tCEC · 2025-07-06

**Clarity:** 4
**Significance:** 3
**Originality:** 3
**Rating:** 5
**Confidence:** 4

**Summary:**

The authors propose a new post-training pipeline to enable "hybrid" thinking in language models. Unlike how existing "thinking" models such as DeepSeek-R1 always employ a long thinking phase, i.e., `<thinking>...</thinking> Final response`, hybrid thinking models adaptive choose whether to employ thinking, or directly respond to the question, i.e., `<no_thinking>Final response</no_thinking>`.

The method employs a two-stage pipeline, similar to that of DeepSeek-R1. The first cold-start SFT phase includes samples that (1) employ thinking for reasoning tasks, and those that (2) provide direct responses to non-reasoning tasks. In the second RL phase, the model is prompted to generate multiple responses in thinking and non-thinking modes, respectively. Positive rewards are given to (1) the mode that achieved the best outcome on average, and (2) individual responses that achieved high reward for the given task.

The method enables higher quality responses across various tasks, and potentially more efficient inference.

**Questions:**

### Questions

- In Table 2 in Appendix C, is WildChat-1M used as the Thinking dataset, and the rest used for Non-Thinking?
- Is the hybrid-thinking model more efficient compared to the thinking model, i.e., by using less output tokens?

### Suggestions

- Some parts of the method explanation could be simplified by using plain words instead of, or supplementing, the mathematical equations. E.g., equation (5) could simply be explained as generating N responses, with `<think>` appended to the prompt for one half, and `<no_think>` to the other.
- (Future work) the method could be improved by enabling the model to reason before making the decision to `<think>` or `<no_think>`. The current training pipeline forces to make this decision in the first decoding step.

### Criteria

I am willing to increase my score if W1 is adequately addressed.

**Ethical Concerns:**

["NO or VERY MINOR ethics concerns only"]

**Final Justification:**

All of my concerns have been addressed

**Limitations:**

Yes

**Quality:**

3

**Strengths And Weaknesses:**

### Strengths

- Hybrid thinking models trained with the proposed method (1) outperform existing thinking models of the same size and (2) outperform baseline hybrid thinking models proposed by the authors, using standard RL algorithms, under equivalent settings as the proposed method.
    - *(W1) However, there is no result on a non-hybrid thinking model under equivalent settings.*
- Analysis shows that the trained model choses between thinking and non-thinking modes adaptively according to the task. Strong evidence of adaptive thinking is shown within the training domain (Figure 3(c), 4) and some evidence is shown for out-of-domain settings (Figure 5).
- Extensive ablations on the choice of specific advantage estimators (GRPO, RLOO, Reinforce++) and the margin hyperparameter are conducted. These confirm robustness to the choice of advantage estimator, and confirm that the margin hyperparameter work as expected.
- Each stage of the training methodology, as well as the proposed metric on hybrid thinking, are simple and straightforward.
- The methodology and experimental settings are well-explained in detail.

### Weaknesses

1. The authors do not compare the proposed hybrid thinking model with an equivalent non-hybrid thinking model. The baseline model in Table 1, DeepSeek-R1-Distill-Qwen, may use a different training protocol from that of the proposed hybrid thinking models.
    - This makes it unclear whether hybrid thinking degrades performance in reasoning-intense tasks.

---

> ### Author Rebuttal · Authors · 2025-07-31
>
> We are grateful to the Reviewer for the extensive review. We address your questions point by point below.
>
> > **Q1:** In Table 2 in Appendix C, is WildChat-1M used as the Thinking dataset, and the rest used for Non-Thinking?
>
> **A1**: In Appendix C, Table 2, WildChat-1M is actually used as the Non-Thinking dataset, while the remaining datasets are employed as the Thinking dataset. We apologize for any confusion caused and will clarify this point explicitly in the latest version of our manuscript.
>
> ---
> > **Q2:** Is the hybrid-thinking model more efficient compared to the thinking model, i.e., by using less output tokens?
>
> **A2**: We have compared the output lengths between the proposed NHTMs (hybrid-thinking models), the thinking-only models (NHTMs-Thinking), and the DeepSeek-R1-Distill-Qwen model in **Table 1 & 2**. We find that:
>
> * Across most benchmarks, the hybrid-thinking models consistently produce shorter outputs while maintaining equal or better performance, demonstrating their ability to reduce unnecessary reasoning for simpler queries and thus improve inference efficiency.
> * On the more challenging AIME24 benchmark, the hybrid-thinking models generate longer outputs compared to both the thinking-only and DeepSeek models, as they engage in more elaborate reasoning to solve complex problems. This additional reasoning leads to significantly improved performance—NHTMs achieve the best accuracy at both the 1.5B and 7B model sizes, with the 7B variant outperforming DeepSeek-R1-Distill-Qwen by **8** percentage points.
>
> These results clearly demonstrate that our hybrid-thinking model achieves improved inference efficiency by adaptively reducing unnecessary reasoning steps for simpler queries.
>
>
> ***Table 1**. Output Length Comparison Across Hybrid, Thinking-Only, and DeepSeek-R1-Distill-Qwen Models on Various Benchmarks (1.5B size)*
>
> | **Model**                        | MATH500 | AIME24 | AMC23 | Olympiad | LiveCodeBench | MBPP | AlpacaEval2.0 | ArenaHard2.0 | **Avg.** |
> |----------------------------------|---------|--------|-------|-----------|----------------|------|----------------|---------------|-------------|
> | DeepSeek-R1-Distill-Qwen-1.5B   | 4059    | **13366**  | 9556  | 10921     | 13002          | 4146 | 4090           | 8285          | 8428        |
> | NHTMs-1.5B-Thinking              | 4463    | 13491  | 10093 | 10680     | 12969          | 4863 | 1986           | 9286          | 8479        |
> | NHTMs-1.5B                       | **3722**    | 13491  | **9065**  | **9490**      | **9342**           | **3103** | **1250**           | **5289**          | **6844**        |
>
> ***Table 2**. Output Length Comparison Across Hybrid, Thinking-Only, and DeepSeek-R1-Distill-Qwen Models on Various Benchmarks (7B size)*
>
> | **Model**                        | MATH500 | AIME24 | AMC23 | Olympiad | LiveCodeBench | MBPP | AlpacaEval2.0 | ArenaHard2.0 | **Avg.** |
> |----------------------------------|---------|--------|-------|-----------|----------------|------|----------------|---------------|-------------|
> | DeepSeek-R1-Distill-Qwen-7B     | 3558    | **9488**   | 6255  | 8635      | 11669          | 2824 | 2209           | 5282          | 6240        |
> | NHTMs-7B-Thinking                | 3547    | 11031  | 6328  | 8736      | 11791          | 3076 | 1628           | 6542          | 6585        |
> | NHTMs-7B                         | **2616**    | 11031  | **4976**  | **7540**      | **8432**           | **1906** | **1086**           | **3416**          | **5125**        |
>
> ---
> > **Weaknesses1**: The authors do not compare the proposed hybrid thinking model with an equivalent non-hybrid thinking model. The baseline model in Table 1, DeepSeek-R1-Distill-Qwen, may use a different training protocol from that of the proposed hybrid thinking models.
>
> **A3**: We appreciate the reviewer’s concern and have conducted an additional experiment for a direct comparison. Specifically, we fine-tuned both the 1.5B and 7B Qwen-2.5-Math models on **~1.1M math and code reasoning data in Thinking-style format**, and compared them against our hybrid-trained models under controlled settings. The two model types are defined as follows:
>
> - **PureThink-SFT models:**
>   These models are trained exclusively on ~1.1M math and code **Thinking-style data** using standard supervised fine-tuning. No hybrid formatting or No-Thinking data is involved.
>
> - **HFT (Think Only) models:**
>   These models are trained via our Hybrid Fine-Tuning (HFT) pipeline, which includes both Thinking and No-Thinking data. During evaluation, they are constrained to always generate responses in the Thinking mode.
>
> As shown in **Table 3**, both variants achieve highly similar performance across multiple reasoning benchmarks at both 1.5B and 7B scales. This demonstrates that **hybrid thinking does not degrade performance on reasoning-intensive tasks**, validating the soundness of our hybrid training design.
>
> ***Table 3**. Comparison of Hybrid and Non-Hybrid Models on Reasoning-Intensive Tasks*
>
> | **Model (Training Type)**         | MATH500 | AIME24 | AMC23 | Olympiad | LiveCodeBench | MBPP | MBPP+ | **Avg.** |
> |----------------------------------|---------|--------|-------|-----------|----------------|------|--------|----------|
> | PureThink-SFT-1.5B               | 87.6    | 34.7   | 72.5  | 49.9      | 17.2           | 63.5 | 53.2   | 54.1     |
> | HFT-1.5B (Think Only)            | 87.8    | 32.7   | 75.0  | 50.0      | 16.8           | 62.2 | 52.9   | 53.9     |
> | PureThink-SFT-7B                 | 94.0    | 56.7   | 92.5  | 59.9      | 38.7           | 80.7 | 68.3   | 70.1     |
> | HFT-7B (Think Only)              | 93.6    | 56.7   | 95.0  | 59.7      | 38.4           | 80.3 | 68.9   | 70.4     |
>
> ---
>
> > **Suggestions1:** Some parts of the method explanation could be simplified to help readers better understand
>
> **A4**: We sincerely thank the reviewer for this valuable suggestion. We fully agree that simplifying parts of the method explanation, such as equation (5), using plain language can significantly improve reader understanding. As suggested, we will revise equation (5) and other parts of the method explanation in the latest version to improve clarity and help readers better understand our method.
>
> ---
> > **Suggestions2:** (Future work) Consider allowing the model to reason before deciding on  or <no_think> instead of making the decision at the first decoding step
>
> **A5**: We appreciate the insightful idea! Indeed, enabling the model to first reason about the context and complexity of the query could potentially enhance the accuracy of selecting the appropriate thinking mode, thereby improving both the overall performance and efficiency. We plan to explore this approach in future work to further improve our hybrid thinking model. Meanwhile, we will include a discussion of this potential enhancement in the latest version of the manuscript.
>
> ----
> We sincerely thank the Reviewer tCEC for the thoughtful comments and constructive suggestions :). Your insights have been invaluable in improving the clarity and completeness of our work. In response, we will incorporate the additional comparisons, clarify the dataset usage, simplify the method description, and include a discussion of this direction in the future work section of the latest version of our manuscript.
>
> If you have any further questions or concerns, please feel free to contact us at any time. We are always available and look forward to further discussions with you. :)
>
> Best regards,
>
> All Authors

---

> > ### Comment · Reviewer_tCEC · 2025-08-01
> >
> > Thank you for the detailed response and for running the additional experiments under the tight time constraints.
> >
> > All of my concerns have been very satisfactorily addressed.
> >
> > The additional results on reasoning lengths shows that the models save more tokens on seemingly simpler tasks, nicely demonstrating the adaptive nature of the method.

---

> ### Author Response · Authors · 2025-08-02
> **Official Comment by Authors**
>
> We sincerely appreciate your thoughtful response and kind recognition of our additional experiments. We are very glad to hear that all your concerns have been satisfactorily addressed.
>
> As suggested, we will incorporate the supplementary experiments and analyses you recommended into the latest version of our manuscript.
>
> Thank you again for your valuable insights and positive feedback.
>
> Best regards,
>
> All Authors

---

### Author Response · Authors · 2025-08-09
**Summary of the Rebuttal Period for ACs and All Reviewers**

Dear Reviewers and ACs,

As the discussion period is almost over, we would like to sincerely thank reviewers and ACs for their efforts in reviewing our paper and for the timely responses during the discussion period.

### **Summary**

> We are encouraged that the reviewers consistently recognize the strengths of our work across novelty, methodology, experiments, and presentation:
>
> *  **Novel and Well-Motivated**: The paper addresses the important and practical problem of “overthinking” in LRMs by introducing a novel and well-motivated hybrid thinking approach. Reviewers noted the idea is "reasonable" and effectively tackles a key efficiency challenge in reasoning models. (Reviewer tCEC, jtLz)
>
> * **Methodologically Sound and Simple**: The proposed two-stage training pipeline (HFT + HGPO) is described as simple, well-designed, and straightforward (Reviewer tCEC, ofyf). The adaptive mode selection is clearly demonstrated, with strong in-domain evidence and some out-of-domain generalization. Robustness is further supported by extensive ablation studies and detailed methodological explanations. (Reviewer tCEC)
>
> * **Experimentally Convincing**: The experiments are solid, comprehensive, and convincing (Reviewer tCEC, ofyf), with NHTMs outperforming strong baselines—including standard LLMs, LRMs, and hybrid variants—on both reasoning and general-purpose benchmarks across multiple model scales. (Reviewer tCEC, ofyf, hw6N)
>
> * **Clearly Presented**: The paper is well-written and easy to follow. (Reviewer tCEC, jtLz, ofyf)
>
> We are glad to see that the **concerns raised by Reviewers tCEC, ofyf, and hw6N have been fully addressed**. Additionally, **Reviewer jtLz has confirmed that most of their concerns have been resolved**. Specifically:
>
> * Reviewer tCEC indicated they would **increase their score** upon seeing the addressed concern (W1).
>
> * Reviewer hw6N expressed that they will **raise their score** following the resolution of their concerns.
>
> * Reviewer jtLz expressed that they will **raise their confidence score and support its acceptance.**
>
> We have meticulously addressed each of the reviewers' suggestions one by one and incorporated the feedback into the revised version. During the rebuttal, we conducted numerous additional experiments and clarifications, all further supporting our conclusions. The key modifications and experiments made during the rebuttal period are summarized below:

### **Additional Experiments:**

>* **Non-Hybrid Model Comparison:** We conducted ablation experiments with non-hybrid models, confirming that hybrid training does not degrade reasoning ability. (Reviewer tCEC W1)
>* **CoT-Compression Comparison:** We compared output token lengths with leading **CoT compression methods** (O1-Pruner, CoT-Valve, DAST), showing NHTMs achieve superior efficiency while maintaining strong accuracy. (Reviewer tCEC Q2, jtLz W2, ofyf W2)
>* **Isolating HGPO's Contribution:** We added controlled RL experiments, **comparing HGPO with standard GRPO under identical training setups**, to isolate the benefit of our adaptive objective. (Reviewer jtLz W1)
>* **Validation of Hybrid Accuracy:** We validated Hybrid Accuracy through **GPT-4o and human evaluations**, confirming its reliability and fairness. (jtLz W3)
>* **Expanded Benchmark Evaluations:** We added evaluations on **GPQA-Diamond, HumanEval, and HumanEval++** for broader LRM comparison. (Reviewer hw6N Q2)
>* **DeepScaleR-1.5B-Preview Comparison:** We compared with **DeepScaleR-1.5B-Preview** (Reviewer hw6N W1)

### **Clarifications:**
>* **Adaptive Behavior Analysis:** We clarified the thinking ratio and token length changes on MATH500 and AIME24, showing why adaptive thinking improves efficiency on simpler tasks. (Reviewer jtLz, Q1)
>* **Methodological Explanation:** We clarified and simplified our methodological exposition, especially regarding the inter-group advantage term in Equation (10), making explicit how the model learns to select the reasoning mode through cross-mode reward contrast. (Reviewers tCEC Suggestion1; ofyf Q1)
>* **Hybrid Accuracy Metric:** We clarified that Hybrid Accuracy measures a model’s internal decision quality rather than cross-model comparison. (Reviewer ofyf W3)
>* **Data Decontamination:** We provided a detailed account of our data decontamination process. (Reviewer hw6N W3)
>* **Future Work Discussion:** We added a discussion on future extensions, such as allowing the model to reason before selecting the mode. (Reviewer tCEC Suggestion2)
>* **General Paper Improvements:** We clarified HFT dataset usage, corrected the missing y-axis label in Figure 5, and fixed grammatical errors (Reviewers tCEC Q1; ofyf W1; hw6N W4)

We would like to express our deepest gratitude once again to the reviewers for their constructive feedback. We believe these extensive experiments and revisions have significantly strengthened our paper and thoroughly addressed the concerns raised.

Best regards,

All Authors

---

### Decision · Program_Chairs · 2025-09-17

**Decision:**

Accept (poster)

**Comment:**

This was a consensus accept. Please take into account how the reviewers initially interpreted the baselines when preparing the final version.